



# STEMMUS-MODFLOW v1.0.0: Integrated Understanding of Soil Water and Groundwater Flow Processes: Case Study of the Maqu Catchment, north-eastern Tibetan Plateau

Lianyu Yu[1, 2, 6], Yijian Zeng[2], Huanjie Cai[1, 6], Mengna Li[2, 3], Yuanyuan Zha[4], Jicai Zeng[5], Hui Qian[3], Zhongbo Su[2, 3]

[1] College of Water Resources and Architectural Engineering, Northwest Agriculture and Forestry University, Yangling, China

10     [2] Faculty of Geo-Information Science and Earth Observation, University of Twente, Enschede, the Netherlands.

[3] Key Laboratory of Subsurface Hydrology and Ecological Effect in Arid Region of Ministry of Education, School of Water and Environment, Chang'an University, Xi'an, China

[4] State Key Laboratory of Water Resources and Hydropower Engineering Sciences, Wuhan

15     University, Wuhan, China

[5] Department of Hydrology and Atmospheric Sciences, University of Arizona, Tucson, USA

[6] Key Laboratory of Agricultural Soil and Water Engineering in Arid Area of Ministry of Education, Northwest Agriculture and Forestry University, Yangling, China.

*Correspondence to*: Zhongbo Su (z.su@utwente.nl), Yijian Zeng (y.zeng@utwente.nl)





**Abstract**

How to efficiently and physically integrate the soil water dynamics with groundwater flow processes has drawn much attention. We present a coupled soil water-groundwater model, considering the two-way feedback coupling scheme, and verified its performance using two synthetic cases (using the fully 3D variably saturated flow (VSF) model simulations as the 'reference') and one real catchment case (using the groundwater table depth and soil moisture profile measurements). By the cross-validation between the observations and various model simulations, the two-way coupling approach is proven physically accurate and is applicable for large-scale groundwater flow problems. Compared to the simulation by groundwater model alone (i.e., only MODFLOW), the coupling of MODFLOW with one soil column reduced the overestimation of groundwater table simulation (taking the VSF model simulations as the reference). The results were further improved as more soil columns were used to represent the heterogeneous soil water-groundwater interactions. Compared to the HYDRUS-MODFLOW, the two-way coupling approach produces a similar spatial distribution of hydraulic heads while better performs in mimicking the temporal dynamics of groundwater table depth and soil moisture profiles. We attribute the better performance to the different coupling strategies across the soil-water and groundwater interface. It is thus suggested to adopt the two-way feedback coupling scheme, together with the moving phreatic boundary and multi-scale water balance analysis, to maintain physical consistency and reduce coupling errors. The realistic implementation of the vadose zone processes (with STEMMUS), coupling approach, and spatiotemporal heterogeneity of soil water-groundwater interactions were demonstrated critical to accurately represent an integrated soil water-groundwater system. The developed STEMMUS-MODFLOW model can be further equipped with different complexities of soil physics (e.g., coupled soil water and heat transfer, freeze-thaw, airflow processes), surface hydrology (snowfall, runoff), soil and plant biogeochemical processes, towards an integrated "from bedrock to atmosphere" modeling framework.



## 1. Introduction

Vadose zone water flow and groundwater flow interact by exchanging water and energy fluxes, making the process of these two subzones closely coupled. Current earth system models (ESMs), land surface models (LSMs), and soil water models (SWMs), devote many efforts in representing soil water and heat dynamics, while the detailed groundwater flow processes, for instance the lateral groundwater flow, is often ignored (Famiglietti and Wood, 1994; Liang et al., 2003; Best, 2011; Niu et al., 2011; Brunner and Simmons, 2012; Milly et al., 2014; Vergnes et al., 2014; de Graaf et al., 2017; Lawrence et al., 2019).

The integrated soil water-groundwater (SW-GW) modelling facilitates the process understanding of the hydrological/ecological system and how it will respond to the future climate changes (Kurylyk et al., 2014a; Kurylyk et al., 2014b; Barthel and Banzhaf, 2016; Zipper et al., 2017). It is particularly important for the integrated water resource management at the regional scale ($10^2$ to $10^5$ km$^2$) (Barthel and Banzhaf, 2016). Furthermore, the integrated SW-GW modelling can be employed to assess the sustainability of the regional ecosystem service and its resilience to climate extremes (Booth et al., 2016; Qiu et al., 2018a; Qiu et al., 2018b; Qiu et al., 2019; Brelsford et al., 2020).

There are quite a number of SW-GW researches conducted in recent years (Maxwell and Miller, 2005; Twarakavi et al., 2008; Niu et al., 2011; Tian et al., 2012; Niu et al., 2014; Bisht et al., 2017). Several approaches have been proposed to integrate the vadose zone processes with the groundwater system, including the conceptual water budget approach with empirical parameterizations (Harbaugh et al., 2000; Scanlon et al., 2002), simplification of vadose zone processes (e.g., the UZF package for MODFLOW, Niswonger et al., 2006), the fully 3D Richards solution (Thoms et al., 2006; Maxwell et al., 2017), and the quasi-3D unsaturated-saturated coupling scheme (Seo et al., 2007; Kuznetsov et al., 2012; Zhu et al., 2012; Mao et al., 2019; Zeng et al., 2019).

Fully 3D Richards solution is physically accurate in describing variably saturated water flow process with a solid theoretical foundation. However, it is computationally expensive and not easy to achieve the numerical convergency for large scale modeling under complex meteorological and hydro-geophysical conditions (Twarakavi et al., 2008). The tipping bucket method, in contrast, requires significantly less computational effort to obtain a converged simulation as it simply considers the vadose zone effect as a certain amount of water flux recharging/discharging the groundwater system (Harbaugh et al., 2000; Scanlon et al., 2002; Twarakavi et al., 2008). This method has been successfully applied in many large-scale groundwater simulations, while additional calibration efforts are necessary as it oversimplifies the vadose zone process. The kinematic wave equation has been used as an alternative to characterize the vadose zone process (Smith, 1983; Smith and Hebbert, 1983). It well captures the gravity-induced water flow while neglects the capillary pressure-induced water flow. In such way, the application of this simple method is case dependent (most applicable in regions with deep water tables and less suitable for aquifers with shallow water tables) (Zha et al., 2019). For the quasi-3D unsaturated-saturated coupling approach,





water flow in the unsaturated zone is assumed only in vertical direction with no lateral flux. The regional scale vadose zone is characterized by multiple soil columns solving Richards equation independently. The saturated zone is solved by the 2D/3D groundwater flow equations. The 1D vadose zone model and 2D/3D groundwater model exchange the boundary information as an integrated model (Seo et al., 2007; Zhu et al., 2011; Mao et al., 2019; Zeng et al., 2019). This quasi-3D coupling scheme simultaneously considers near-

surface meteorological conditions, surface hydrological conditions, water table fluctuations, and the hydrothermal properties of vadose zone. It sustains the physical reality of the unsaturated-saturated zone processes and improves the numerical stability and computational efficiency (Twarakavi et al., 2008; Zeng et al., 2019).

Proved to be a promising way balancing the computational cost and physical accuracy, the quasi-3D

coupling method has drawn much attention and efforts from soil water and groundwater modelers (Havard et al., 1995; Seo et al., 2007; Mao et al., 2019; Zeng et al., 2019). Generally, quasi-3D coupling scheme can be realized by exchanging the groundwater recharge and water table levels across the SW-GW interface. According to (Furman, 2008), such SW-GW coupling is classified into three categories: i) weakly coupled or one-way coupled, which directly adds the soil water flow solutions onto the groundwater models (e.g.,

SVAT-MODFLOW, (Facchi et al., 2004); UZF-MODFLOW, Niswonger et al., 2006; HYDRUS-MODFLOW, Seo et al., 2007; SWAP-MODFLOW, Xu et al., 2012). Although this scheme is easy to implement, it can be problematic due to its failure in capturing the SW-GW interactions (Beegum et al., 2018; Mao et al., 2019; Zeng et al., 2019; Brandhorst et al., 2021). ii) the fully coupling scheme, which simultaneously solves the soil-water and groundwater flows by building the nodal hydraulic connections

and matrices across the interface. It is mathematically rigorous but computationally expensive, thus limits its application in large regional scale problems. iii) two-way feedback coupling scheme, which exchanges water flux or water table levels at the interface nodes. The feedback coupling scheme maintains the consistency of exchanging quantities in space and time, thus effectively avoids the mass balance error accumulation, with moderate computational effort. Nevertheless, the feedback coupling scheme may face

the scale-mismatch problem (the parameterization of the water release at the phreatic surface/shared nodes differs from soil water and groundwater models). This problem can result in numerical instabilities and coupling errors (Zeng et al., 2019; Brandhorst et al., 2021). Multi-scale water balance analysis appears to be a promising way to link the soil water and groundwater quantities at the phreatic surface (Zeng et al., 2019).

Under certain conditions (e.g., high topographic gradient), the lateral water fluxes between saturated regions of soil columns cannot be neglected (Zeng et al., 2019). The moving-boundary strategy, using the adaptive soil column bottom boundary instead of the fixed one in response to the fluctuating groundwater table depth, has been reported to overcome the numerical difficulty caused by such saturated lateral water flow and thus reduce the coupling errors (Zeng et al., 2019).


In this study, we coupled the soil water model (STEMMUS) with the groundwater model (MODFLOW) in
a two-way feedback manner. The iterative feedback scheme, multi-scale water balance analysis, and
moving boundary approach were all incorporated in the coupling framework. Two test cases and one real-
world catchment scale validation were conducted to testify the developed SW-GW coupling model
STEMMUS-MODFLOW. In the following sections, the governing equations, coupling procedure of

STEMMUS-MODFLOW, the model setup of two test cases and one catchment scale case are described in
Section 2. Section 3 presents the validation results, followed by the discussion in Section 4. This work is
summarized in Section 5.

## 2. Methodology

### 2.1 Governing equations

MODFLOW assumes that the ground water is with constant density and the porous medium is
noncompressible, the 3-D groundwater flow is described by the following partial-differential equation
(Harbaugh, 2005).

$$\frac{\partial}{\partial x}\left(K_{xx}\frac{\partial H}{\partial x}\right) + \frac{\partial}{\partial y}\left(K_{yy}\frac{\partial H}{\partial y}\right) + \frac{\partial}{\partial y}\left(K_{zz}\frac{\partial H}{\partial y}\right) + W = S_S\frac{\partial H}{\partial t} \tag{1}$$

where $K_{xx}$, $K_{yy}$, and $K_{zz}$ (m s$^{-1}$) are the hydraulic conductivity along the $x$, $y$, and $z$ coordinate axes; $H$ (m) is
the hydraulic head; $W$ (s$^{-1}$) is the volumetric flux per unit volume representing sources and/or sinks of water;

$S_s$ (m$^{-1}$) is the specific storage of the porous material; $t$ (s) is time.

The vadose zone water flow is mainly driven by both the gravity and capillary force, the vapor transport
mechanisms, i.e., diffusion and advection are included here. The governing equation can be mathematically
described using the modified Richards equation (Milly, 1982; Zeng et al., 2011a).

$$\frac{\partial}{\partial t}(\rho_L\theta_L + \rho_V\theta_V) = -\frac{\partial}{\partial z}(q_{Lh} + q_{LT} + q_{Vh} + q_{VT}) - S$$

$$= \rho_L\frac{\partial}{\partial z}\left[K\left(\frac{\partial\psi}{\partial z} + 1\right) + D_{TD}\frac{\partial T}{\partial z}\right] + \frac{\partial}{\partial z}\left[D_{Vh}\frac{\partial\psi}{\partial z} + D_{VT}\frac{\partial T}{\partial z}\right] - S \tag{2}$$

where $\rho_L$ and $\rho_V$ (kg m$^{-3}$) are the density of water and water vapor, respectively; $\theta_L$ and $\theta_V$(m$^3$ m$^{-3}$) are the

soil liquid and vapor volumetric water content, respectively; $q_{Lh}$ and $q_{LT}$ (kg m$^{-2}$ s$^{-1}$) are the soil liquid
water flow driven by the gradient of soil matric potential $\frac{\partial\psi}{\partial z}$ and temperature $\frac{\partial T}{\partial z}$, respectively. $q_{Vh}$ and $q_{VT}$
(kg m$^{-2}$ s$^{-1}$) are the soil water vapor fluxes driven by the gradient of soil matric potential $\frac{\partial\psi}{\partial z}$ and temperature
$\frac{\partial T}{\partial z}$, respectively. $D_{TD}$ (kg m$^{-1}$ s$^{-1}$ K$^{-1}$) is the transport coefficient of the adsorbed liquid flow due to





temperature gradient; $D_{Vh}$ (kg m$^{-2}$ s$^{-1}$) is the isothermal vapor conductivity; and $D_{VT}$ (kg m$^{-1}$ s$^{-1}$ K$^{-1}$) is the thermal vapor diffusion coefficient.

STEMMUS takes into account different heat transfer mechanisms, including heat conduction ($\lambda_{eff}\frac{\partial T}{\partial z}$), convective heat transferred by liquid and vapor flow, the latent heat of vaporization ($\rho_V\theta_V L_0$), and a source term associated with the exothermic process of wetting of a porous medium (integral heat of wetting) ($-\rho_L H_W \frac{\partial \theta_L}{\partial t}$).

$$\frac{\partial}{\partial t}\left[(\rho_s\theta_s C_s + \rho_L\theta_L C_L + \rho_V\theta_V C_V)(T - T_{ref}) + \rho_V\theta_V L_0\right] - \rho_L H_W \frac{\partial \theta_L}{\partial t}$$
$$= \frac{\partial}{\partial z}\left(\lambda_{eff}\frac{\partial T}{\partial z}\right) - \frac{\partial}{\partial z}[q_L C_L(T - T_{ref}) + q_V(L_0 + C_V(T - T_{ref}))] - C_L S(T - T_{ref}) \tag{3}$$

where $\rho_s$ (kg m$^{-3}$) is the soil solids density; $\theta_s$ is the volumetric fraction of solids in the soil; $C_s$, $C_L$, and $C_V$ (J kg$^{-1}$ K$^{-1}$) are the specific heat capacities of soil solids, liquid, and water vapor, respectively; $T_{ref}$ (K) is the arbitrary reference temperature; $L_0$ (J kg$^{-1}$) is the latent heat of vaporization of water at the reference temperature; $L_f$ (J kg$^{-1}$) is the latent heat of fusion; $H_W$ (J kg$^{-1}$) is the differential heat of wetting (expressed as the amount of heat released when a small amount of free water is added to the soil matrix). $q_L$ and $q_V$ (kg m$^{-2}$ s$^{-1}$) are the liquid and vapor water flux, respectively. Additional details on the equations for solving the coupled water and heat equations can be found in (Zeng et al., 2011b, a; Zeng and Su, 2013; Yu et al., 2018).

## 2.2 Coupling procedure

### 2.2.1 General framework of STEMMUS-MODFLOW

The current coupling method is through exchanging the boundary information between soil water model STEMMUS and groundwater model MODFLOW (Figure 1). Both STEMMUS and MODFLOW model are run separately. First, they run the preparation and initialization part independently. Once MODFLOW starts a new time step simulation, MODFLOW will inform STEMMUS with the time information (absolute time, time step) and the updated water pressure. Then STEMMUS model is told to start the simulation within a given time step. The received water pressure $H^N$ are set as the bottom boundary conditions for the STEMMUS model. After a certain number of iterations, STEMMUS model will converge to successfully update the state variables ($\psi, T$) and boundary fluxes ($Q$). The updated bottom boundary flux is sent back to MODFLOW model as its groundwater recharge. MODFLOW will formulate the matrix equations and iteratively solve for state variables $H^{N+1}$ (updated ground water head) for the current time step. Such loop will continue till the end of simulation period.

### 2.2.2 Spatial coupling


Generally, the user can divide the study area by making as many unsaturated profiles as he likes. In the following example (Figure 2a), two kind of soil profiles were employed to represent the hydraulic and thermal process of the vadose zone of the study area (i.e., STEMMUS soil profile 1 and 2). Each soil profile was associated with a group of MODFLOW cells. The depth to groundwater in each cell of the zone is averaged to determine the hydraulic head at the bottom of the corresponding soil profile. The flux from the bottom of soil profile is then applied to each cell of the zone as the groundwater recharge for the time step. The iterative feedback coupling scheme is utilized to have both the soil water and groundwater models converged at the coupling interface (see Appendix A). To overcome the issue of the lateral flux in the saturated part of soil columns, the moving lower boundary approach for soil water model is adopted here. The soil column is resized after each MODFLOW time step, adjusting the lower boundary according to the updated groundwater levels (see details in Appendix A).

### 2.2.3 Temporal coupling

Usually, the time step of groundwater models is larger than that of soil water models (Figure 2b). MODFLOW is operated with a prescribed time step ($10^3$-$10^6$s). Within the given time step, STEMMUS model will adapt its own time step (variable between $10^0$-$10^3$s) to the converged simulation results. As soon as the accumulative simulation time of STEMMUS reaches that of MODFLOW, STEMMUS will relax and send the bottom boundary information to MODFLOW for the corresponding time step.

### 2.2.4 Multi-scale interface water balance analysis

To minimize the mass error at the interface, the multi-scale water balance analysis is conducted. The space- and time-splitting strategy is used to estimate water budgets of soil water and groundwater models at respective scales. For the soil water zone affected by the fluctuation of groundwater table, specific yield for the small scale (soil water model) is calculated by vertically integrating the soil water storage. Large scale specific yield quantifies the water release of the phreatic aquifer. By scale matching of the water budget components at the interface of respective scales, water balance is conserved and the upper boundary flux for the groundwater flow model can be achieved. In such way, the dynamic changing of specific yield at both scales is physically maintained and linked (see Appendix A for detail; Zeng et al., 2019).

### 2.3 Two test cases

Two test cases were used to verify the developed SW-GW coupling model. The true reference was obtained by running the fully 3D variably saturated flow model (VSF). MODFLOW models without and with STEMMUS model were run to evaluate the impact of coupling schemes on SW-GW interactions.

**Case 2D:** large groundwater drawdown

This case was used to test the model performance under the condition of large groundwater drawdowns (see the schematic plot of the cross section in Figure 3). The 2D domain dimension is 5000 cm × 1000 cm. The


boundary conditions (top, bottom, left, and right) are all set as non-flux boundary. The initial hydrostatic head of the cross section is 700 cm. Two pumping wells are applied as hydraulic stresses to the cross-section field. Well 1 is added at $x = 2500$ cm with a pumping rate $Q_1 = 2\times10^4$ cm$^2$ d$^{-1}$ per unit width. Well 2 is added at $x = 5000$ cm with a pumping rate $Q_2 = 1\times10^4$ cm$^2$ d$^{-1}$ per unit width. The pumping screen for both wells are at $z = 0$–200 cm. Soil type of loam is used, and the hydraulic properties is listed in Table A1.

The total simulation period lasts 50 days. As for the spatial discretization, MODFLOW as well as the reference VSF model, uniformly divided the cross section along $x$ direction into 100 columns with the width of 50 cm. The vertical direction is discretized into 91 layers, with the thickness ranging from 5 cm to 200 cm (top to bottom). For the 1D vadose zone model STEMMUS, the cross section along the $z$ direction is discretized evenly into 1000 layers with the thickness of 1 cm. Number of 1D soil columns is designed as

1, 5, and 11 columns, all evenly distributed along $x$ coordinate.

**Case 3D:** pumping and irrigation

More complex conditions with various stresses are equipped in this 3D case to verify the coupled STEMMUS-MODFLOW model performance. A phreatic aquifer with domain of 1000 m × 1000 m × 20 m is stressed by constant irrigation and pumping wells (Figure 4). The subdomains $(x, y) = (0$–440 and 560–

1000 m) are recharged at a rate of 3 mm d$^{-1}$. For subdomains $(x, y) = (560$–1000 and 0–440 m), the infiltration rate is 5 mm d$^{-1}$. Three pumping wells with a constant pumping rate of 30 m$^3$ d$^{-1}$ are set up. The screens are located at $(x, y,$ and $z) = (220$ m, 220 m, and 5–10 m), (500 m, 500 m, and 5–10 m), and (780 m, 780 m, and 5–10 m) for Well no. 1, 2, and 3, respectively. The bottom and lateral boundaries are non-flux. The initial hydrostatic head of the aquifer is 18 m. The used soil parameters are given in Table A1.

The total simulation period lasts 60 days. The aquifer is uniformly discretized into 25 × 25 cells with the width of 40 × 40 m. The vertical discretization is increasingly finer from bottom to the top with the thickness ranging from 2 m to 0.1 m. For the 1D soil profile, the discretization is 0.1 m × 30 and 0.4 m × 5. The zonation is designed differently in terms of soil column numbers and their locations (Figure 4).

**2.4 Catchment scale SW-GW interactions**

**2.4.1 Maqu catchment and model setup**

The STEMMUS-MODFLOW model was used to investigate the SW-GW interactions in Maqu catchment, which is a typical cold climate with dry winter and warm summer. The annual average temperature is about 1.8 °C. The annual precipitation is about 620 mm. The selected Maqu catchment is about 38.86 km wide from east to west and 26.7 km long from north to south. The domain size is about 536 km$^2$. Ground surface

elevation decreases from 4017 m to 3367 m from the northwest to east. Field campaigns were conducted over Maqu in 2017 and 2018, including lithology survey, water table levels, slug test, magnetic resonance sounding (MRS) and time-lapse electrical resistivity tomography (ERT) measurements (Li et al., 2021). It provides us with detailed hydro-geophysical information of the study area.





The related data includes the specific 3D domain dimensions of the catchment, and associated soil
properties, meteorology forcing data, groundwater table depth measurements, as well as hydro-geophysical
conditions. According to the field survey and campaign of geomorphology and geology, the selected area
can be divided into two parts, the western mountainous area and flat eastern area. The sediments are
alluvial deposits with intercalated eolian units in eastern flat area. Soil texture is finer at topsoil layers
(sandy loam) and coarse at deep soil layers (sand with gravel). The western mountains are feldspathic
quartzose sandstone and sandy slate with soil covered at the top (Li et al., 2021). The western boundary, i.e.,
mountain divide, is the well-defined hydrogeological borders, to which no-flow boundary was applied.
Yellow river flows along the eastern boundary, which was set as the Time-Variant Specified-Head
boundary (MODFLOW CHD package). The northern and southern segments were assumed as the no-flow
boundaries.

Spatial variation of groundwater table depth was collected in August 2018. Long term simulations (1979–
2018) were run for approaching the steady state, to calibrate the hydraulic parameters. The precipitation,
from the China Meteorological Forcing Dataset (CMFD, He et al., 2020) with spatial resolution of 0.1
degree, was applied to the domain. Precipitation infiltration factors are in the range of 0.05–0.15, which is
lower in the mountainous area with steep terrain and larger in the flat area. Potential evapotranspiration $ET_0$
was from ERA5 dataset (spatial resolution of 0.1 degree). The unconfined aquifer was divided into 5 layers
for numerical simulation. The bottom of aquifer was set according to the bedrock depth dataset (Yan et al.,
2020). The domain is divided into uniform grids of 500 m × 500 m. The top surface and bedrock elevations
are presented in Figures 6a & b. Based on the pumping test, slug tests, geophysical exploration campaign
and knowledge, the horizontal hydraulic conductivity $K_h$ was initially assigned, varied from 0.01 m d$^{-1}$ to 5
m d$^{-1}$. The vertical hydraulic conductivity $K_v$ was assigned as a ratio of $K_h$, i.e., 0.1*$K_h$. The specific yield
was assigned uniformly as 0.15. River channel network is obtained from the local field survey and verified
against the land surface DEM. MODFLOW River package was used to interpret the river-groundwater
interactions.

### 2.4.2 Model calibration

The aim of the calibration process is to obtain the proper initial groundwater head condition and hydraulic
parameters. Long-term transient model (REC-ET for MODFLOW) was developed, driven by the annual
average precipitation, potential evapotranspiration (from 1 January 1979 to 31 December 2018). The whole
simulation period was divided into 40 stress periods, each with 12-time steps (i.e., monthly). The model
was calibrated manually in a forward mode. We started the calibration process by first adjusting the
initially assigned hydraulic conductivity value and its zonation (K-zones). The hydraulic conductance of the
riverbed conductance was then slightly changed. The calibration target is to minimize the difference
between simulations and observations of groundwater table elevations and at the meantime to be consistent
with the hydro-geological conditions as surveyed. It is to note that the observations were mainly collected
from the eastern flat area, the calibration process was focused on the eastern part.



### 2.4.3 Coupled model simulations

After the optimization of groundwater flow model, the STEMMUS-MODFLOW model with a finer temporal resolution from 1 January 2016 to 5 August 2018 was run. The top surface and bedrock elevations are presented in Figures 5a & b. The subzones together with the current available measurements of groundwater table depth and soil moisture profile is shown in Figure 5c. The whole domain was discretized into 44 subzones for running soil model STEMMUS, with the soil vertical discretization thickness from 2 m to 0.01 m (finer on topsoil layers). Nine soil water content profile monitoring points were setup in this region in 2011 (Su et al., 2011; Dente et al., 2012) with 4 currently available on 5 August 2018 (Figure 5c). The meteorological forcing (potential evapotranspiration $ET_0$ and precipitation P) was shown in Figure 5d.

## 3. Results

### 3.1 Case 2D

There are two drawdowns corresponding to the pumping well locations (Figure 6). Compared with that estimated by VSF model, the STEMMUS-MODFLOW model with 5 and 11 soil columns well reproduced the amplitude of drawdowns, while underestimated by model with one soil column (Figures 6a &b). Both the water tables and head solutions were overestimated using MODFLOW alone, indicating the important role of vadose zone process and its interactions with groundwater dynamics (Figures 6a &b).

### 3.2 Case 3D

Figure 7 shows the simulated water table at A-A' cross section for Case 3D. From the fully 3D VSF model simulations, three pumping wells with the same pumping rate resulted in three different water table positions, lower water table for low-infiltration zone while higher water table for high-infiltration zone. The STEMMUS-MODFLOW with 9 and 16 subzones produced the similar variation trend. However, MODFLOW alone (without coupled with vadose zone model) produced the higher water table at three pumping well locations. The STEMMUS-MODFLOW-simulated spatial patterns of the phreatic surface solution agreed with that from VSF model simulations (Figure 8), although some slight deviations can be seen. MODFLOW alone, however, cannot reproduced the same spatial pattern in the two zones with infiltration and pumping.

### 3.3 Maqu catchment simulation

### 3.3.1 Calibration results (1979-2018)

By tuning the hydraulic conductivity zones, values (the calibrated horizontal hydraulic conductivity $K_h$ varied from 8e-4 m d$^{-1}$ to 470 m d$^{-1}$), river conductance, we obtained the acceptable model results as shown in Figure 9. The range of the difference between the simulated and observed heads is within 0.5 m.





MODFLOW-simulated water heads were highly correlated to the observed ones with $R^2$ of 0.9996 and root mean square errors (RMSE) of 0.138 m (Figure 9 and Table 1). MODFLOW with the tuned hydraulic parameters well mimicked the spatial distribution of groundwater table depth. The calibration results were acceptable and the hydraulic parameters can be further used for the model intercomparison.

### 3.3.2 Intercomparison results

**1) Spatial variation of groundwater table**

Table 1 shows the statistical performance of used models in producing the regional water table elevations. Slightly worse than the calibration results, all models (MODFLOW, HYDRUS-MODFLOW and the coupled STEMMUS-MODFLOW) can well simulate the spatial distribution of the water table levels (the coefficient of determination are 0.9987, 0.9811 and 0.9727, MAEs are 0.203, 0.754 and 0.631 m, RMSEs are 0.15, 0.95 and 0.86 m for MODFLOW, HYDRUS-MODFLOW and the STEMMUS-MODFLOW, respectively).

Figure 10 presents the spatial variations of hydraulic head elevations estimated by MODFLOW and the STEMMUS-MODFLOW in August 2018. The hydraulic head elevations were simulated higher in the western mountain area, while lower in the eastern region. Water heads became lower approaching the river segments, indicating that groundwater flows toward the river. Two models produced the similar spatial patterns of hydraulic heads.

**2) Time series of groundwater table**

One groundwater monitoring well was installed on 19 November 2017. The observed groundwater table levels were used to validate the model performance in reproducing the temporal dynamics of groundwater table depth (Figure 11). There is an increasing trend with time in the estimated groundwater table elevations by MODFLOW-Only and STEMMUS-MODFLOW models. The daily MODFLOW simulations show a seasonal fluctuation along the monthly simulation values with the maximums occurring during the September and minimums occurring in May. This corresponds to the seasonal patterns of $ET_0$, i.e., larger values in May and smaller values in September. The observed groundwater levels, however, present a delayed maximum value, which may be induced by a pumping test conducted on 19 November 2017. Such water abstraction results in a drawdown of groundwater levels, and then groundwater from the vicinity flows towards the pumping well and then recovers after the pumping test. Together with the recharge from precipitation, groundwater levels approach its maximum in February 2018. Groundwater level fluctuations estimated by MODFLOW at daily time scale agreed with the observed ones after May 2018, while there is an overestimation from MODFLOW simulations.

Compared to the MODFLOW estimated groundwater table elevations, STEMMUS-MODFLOW produced lower groundwater table levels and lower seasonal fluctuations. It indicates a weaken precipitation recharging effect estimated from the STEMMUS-MODFLOW. After May 2018, groundwater levels from



the STEMMUS-MODFLOW agreed well with the observations. The groundwater table elevation simulations from HYDRUS-MODFLOW, however, present a decreasing trend, which indicates that the effect of vadose zone was overestimated.

**3) Soil moisture profile**

The development of soil water profile is influenced by both the soil water and groundwater processes (Figure 12). The initial soil moisture profile is obtained from the assumed hydrostatic hydraulic head profile. Soil water content was low at the top surface layers, nonlinearly increased till the phreatic surface then remained saturated.

HYDRUS simulations without MODFLOW were ran for model intercomparison in terms of unsaturated soil moisture profile. The phreatic surface remained unchanged from the HYDRUS simulations with zero bottom water flux (BC = 0 m s$^{-1}$). HYDRUS simulated soil moisture profile started to differ from the initial hydrostatic state at the soil depth of about 2 m. Higher than 2 m in the profile, HYDRUS simulated soil moisture increased till the depth of 0.01 m. This zone is more influenced by the surface meteorological forcing (P, ET$_0$), termed soil water (SW) zone here. It is affected by the surface hydrometeorological conditions, and vadose zone soil hydraulic properties. For the HYDRUS simulations with water recharging (BC = 1E-8 m s$^{-1}$), groundwater table depth moved upwards. In contrast, groundwater level moved downwards given that water is flowing into the soil bottom boundary (BC = -1E-8 m s$^{-1}$). There is no difference in the simulated unsaturated soil moisture between HYDRUS simulations with and without water recharge/discharge (BC = 0 and BC = ±1E-8 m s$^{-1}$). The governing water transport process for the soil depth below the groundwater table depth is mainly the groundwater flow, termed groundwater (GW) zone here (Figure 12e). In-between the SW and GW zones, soil moisture remains unchanged comparing to the initial hydrostatic state, termed deep soil zone (DS) here.

Compared to the position of groundwater table depth at the initial time (t = 0), the HYDRUS-MODFLOW and STEMMUS-MODFLOW estimated end-time groundwater table depth was significantly shallow for subzone 1#, 7#, and 41#, while remained relatively steady for subzone 16#, and 32#. It indicates that soil columns of the subzone 1#, 7#, and 41# are gaining water. Such amount of water increased deep zone soil moisture and even away from the initial hydrostatic conditions as the increase of recharging water (from subzone 1# to 7# and 41# with the gradient of elevation decreases, GW zone increases while DS zone decreases and even disappears). Moving to upper soil layers, the soil moisture profile approaches the HYDRUS simulations as surface meteorological forcing is more dominant. Compared to the HYDRUS simulations, the SW zone becomes shallower as the decrease of groundwater table depth, which means that part of the SW zone is also affected by the groundwater table fluctuations (7# and 41#). HYDRUS-MODFLOW presents an increased SW zone for Subzone 1#. With the decrease of groundwater table depth, the SW zone decreased from HYDRUS-MODFLOW simulations (from Subzone 1# to 7# and 41#).



Compared to the measured soil moisture profiles, HYDRUS model agreed well for Subzone 32# while
overestimated for Subzone 16#. The STEMMUS-MODFLOW simulated soil moisture profile was mostly
in the variation range of soil water content measurements for Subzone 16# while underestimated for
Subzone 32#. Compared to the HYDRUS only simulations, the SW zone is losing water from the
STEMMUS-MODFLOW simulations (Figure 12 c & d), which means the groundwater level increase (can
be also seen in Figure 11). The soil moisture was overestimated by the HYDRUS-MODFLOW for Subzone
16# and 32#. SW zone from HYDRUS-MODFLOW simulations is gaining water from the phreatic aquifer
(corresponding to the declining trend of groundwater table elevations in Figure 11). These discrepancies
can be attributed to that the exchange of information across the SW-GW interface is not realistically
delivered by the HYDRUS-MODFLOW (Beegum et al., 2018; Brandhorst et al., 2021).

## 4. Discussion

**4.1 Role of vadose zone processes**

Given precipitation/snowfall, water flow processes include surface/subsurface runoff, infiltration, root
water uptake, and evaporation, and some will eventually drain as groundwater recharge. Considering the
spatiotemporal dynamics, the water flow is highly variable depending on the hydrologic, geologic, and soil
hydrothermal conditions. Thus, it results in a heterogenous groundwater recharge/discharge, i.e., SW-GW
interactions. Without considering the vadose zone, groundwater table will be underestimated/overestimated
as no groundwater recharge/discharge was simulated (see Figure 6). This affects the entire groundwater
system with damping influence as depth goes deeper. Such bias can be alleviated by considering the SW-
GW interactions homogeneously (i.e., one soil column for STEMMUS-MODFLOW). Only by the proper
consideration of the heterogeneous SW-GW interactions, the groundwater table can be realistically
produced (Figure 7). Adding the infiltration process, the situation can be more complex. The soil
infiltration water recharges the groundwater system and enhances the SW-GW interactions (Mao et al.,
2019). The phreatic surface is the dynamic balance between the infiltration and SW-GW interaction
processes. Other than the infiltration process, the heterogeneity of vadose zone flow and SW-GW
interaction process were identified and should be well represented to produce the groundwater flow
dynamics in space and time (Figure 6 & 7). The soil water balance approach (e.g., REC-ET package in
MODFLOW), simply assigning the groundwater surface boundary as the resident of water budget equation,
cannot reproduce the realistic groundwater table dynamics. It is because not only the vadose zone process is
neglected by water budget approach (i.e., no infiltration, no capillary pressure driven flow, (Gee and Hillel,
1988; Twarakavi et al., 2008)), but also the heterogeneity of the vadose zone flow (Mao et al., 2019) and
SW-GW interactions is overlooked (Figure 6 & 7). For the SW-GW coupled modelling development, it is
suggested to be verified not only based on its vadose zone flow process, applicability to different spatial
and temporal scales and various meteorological and climatic conditions (Twarakavi et al., 2008), but also
should take into account the capability in mimicking the dynamic and heterogeneous SW-GW interactions.



### 4.2 The effect of SW-GW coupling approach

To mathematically implement the SW-GW interaction process, various efforts have been made (Seo et al., 2007; Twarakavi et al., 2008; Zhu et al., 2012; Zeng et al., 2019). In this work, three SW-GW coupling approach, i.e., simple coupling (REC-ET packages for MODFLOW), one way coupling (HYDRUS-MODFLOW), and two-way feedback coupling, were intercompared in a cold region catchment. From current available dataset, three methods can well reproduce the spatial variation of groundwater heads

(Table 1). For the temporal dynamics, REC-ET packages for MODFLOW overestimated the groundwater table levels (Figure 11), indicating that the effect of vadose zone flow process is underestimated. HYDRUS-MODFLOW, in contrast, underestimated the groundwater table elevations. As argued by (Beegum et al., 2018), HYDRUS-MODFLOW suffers from the problem that the exchange quantities of the unsaturated and saturated zone are inconsistent, which can result in the sudden inflow or outflow of the

vadose zone and accumulative errors. The other reason can be attributed to the fixed 1D soil column depth, which may produce more lateral groundwater flux into the unsaturated zone model (Seo et al., 2007). The STEMMUS-MODFLOW generally matches the groundwater table depth measurements, which is also found consistent with soil moisture profile measurements (Figure 12). It demonstrates the validity of the coupling method. The adopted coupling method overcomes the scale-mismatch of exchange quantities and

minimize the coupling errors due to the nontrivial lateral water flux across the saturated zones of soil columns (Zeng et al., 2019).

### 4.3 The limitations and outlook

The current work presented a coupled soil water-groundwater model and verified its performance using two test cases (using the fully 3D VSF model simulations as the truth) and one real world catchment case

(taking groundwater table depth and soil moisture profile measurements as the truth). By the cross validation among the observations, and various models, the STEMMUS-MODFLOW is found physically accurate and applicable in regional scale groundwater problems. In addition, the role of vadose zone processes, groundwater flow, coupling approach, and SW-GW interactions was highlighted.

However, regional groundwater simulations in Maqu are with uncertainties and require more effort to

confirm the simulation results. More datasets are needed to better constrain and validate this reginal groundwater modelling case, including the reliable meteorological forcing ($ET_0$, P) at desirable time and space scales, time series of groundwater table depth measurements, river flow, soil moisture profile measurement, and subsurface hydraulic property profile information.

Furthermore, soil thermal effect is important not only in the vadose zone but also in the groundwater

system under certain conditions. Xie et al. (2021) demonstrated the important role of freezing-induced water migration in shallow groundwater systems under semi-arid climate conditions. Lateral water inflow resulted in the groundwater level rise and further enhanced the freezing-induced groundwater migration.





The permafrost dynamics can alter the groundwater and surface water exchange and groundwater discharge to the surface was reported three-fold increase in northern Tibetan Plateau under the increasing air

temperature scenario, which are mainly temperature driven (Ge et al., 2011; Evans and Ge, 2017). Moreover, temperature has been identified as a useful tracer for inferring groundwater fluxes (Bense and Kurylyk, 2017; Bense et al., 2017; Irvine et al., 2017; Kurylyk et al., 2017; Kurylyk and Irvine, 2019; Bense et al., 2020).

Current SW-GW modelling work seldom considers the thermal effect and lacks real-case applications and

verifications (see Appendix Table A2). The developed SW-GW coupling model in this work will facilitate further the manipulation of sub-models with different complexity of vadose zone physics (thermal flow, soil water and heat coupling transfer, freeze-thaw, airflow processes; Yu et al., 2020b), surface hydrology (snowfall, Yu et al., 2021, runoff, Mastrotheodoros et al., 2020), connection with other relevant processes (soil and plant biogeochemical process, Yu et al., 2020a), towards an integrated groundwater-soil-plant-

atmosphere earth system modelling framework.

## 5. Conclusion

The performance of the coupled soil water and groundwater model is verified using two test cases and the Maqu catchment observatory. Vadose zone process and the heterogeneity of soil water-groundwater interactions is demonstrated important in reproducing the water table fluctuation dynamics. Realistic

zonation and parameterization of unsaturated soil columns helps to enhance the model performance. Compared to HYDRUS-MODFLOW, the STEMMUS-MODFLOW produces a similar spatial distribution of hydraulic heads. However, better performance was found in mimicking the temporal dynamics of groundwater table depth and soil moisture profiles. This improved performance is due to the different coupling strategies across the soil-water and groundwater interface. It is suggested to adopt the moving

phreatic boundary, two-way iterative feedback coupling scheme and multi-scale analysis to maintain the physical rational and reduce the coupling errors. The developed STEMMUS-MODFLOW has demonstrated its applicability and can be further equipped with different complexity of vadose zone physics, surface hydrology, soil and plant biogeochemical process, towards an integrated "from bedrock to atmosphere" modelling framework.



*Code and data availability.* The coupled soil water-groundwater (SW-GW) model STEMMUS-MODFLOW was developed based on STEMMUS (Simultaneous Transfer of Energy, Momentum and Mass in Unsaturated Soils) and MODFLOW (Modular Three-dimensional Finite-difference Ground-waterFlow Model). The original STEMMUS source code is available from the GitHub website via https://github.com/yijianzeng/STEMMUS, licensed under the Apache License, Version 2.0. The
MODFLOW-2005 is publically available from Harbaugh et al., (2017) (http://dx.doi.org/10.5066/F7RF5S7G). The coupled STEMMUS-MODFLOW v1.0.0 code is archived in Zenodo (https://doi.org/10.5281/zenodo.7372483), can be available upon reasonable request from the correspondence authors. The current code is tested on the Ubuntu 18.04.6 LTS (GNU/Linux 4.15.0-173-generic x86_64). The relevant hydrogeological data can be accessed at https://doi.org/10.17026/dans-z6t-
zpn7 (Li et al., 2020a), and the processed ERT, MRS, and TEM data is available at the National Tibetan Plateau Data Center with the link https://doi.org/10.11888/Hydro.tpdc.271221 (Li et al., 2020b).

*Author contribution.* ZS, YJZ, and LY designed and conceptualized this study; YJZ and ZS provided the original version of STEMMUS model; YYZ and JZ provided the HYDRUS package for MODFLOW code;
LY developed the STEMMUS-MODFLOW model; ZS, YJZ, HQ, ML, LY coordinated and carried out the hydrogeological and hydrogeophysical survey in Maqu catchment; LY prepared the original draft of the paper, LY, YJZ, HJ, ML, YYZ, JZ, HQ, and ZS all contributed to the reviewing and editing of the manuscript.

*Competing interests.* The authors declare that they have no conflict of interest.

**Acknowledgment**

This work is supported by the National Natural Science Foundation of China (grant no. 41971033) and supported by the Fundamental Research Funds for the Central Universities, CHD (grant no.
300102298307). The authors would like to thank the editors and referees for their helpful comments and suggestions on improving the manuscript.



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





## Appendix A

**A1. Multi scale water balance analysis**

Different spatiotemporal scales are operated for soil water and groundwater models. Soil water models are running at $\Delta z = 10^{-3} - 10^0$ m and $\Delta t = 10^0 - 10^3$ s, while for groundwater models, the domain is usually discretized at $\Delta x/y = 10^0 - 10^3$ m and $\Delta t = 10^3 - 10^6$ s.

Within the large-scale time step $\Delta T = T^{J+1} - T^J$, and local area of interest $\overline{M} = z_s - z_0$ (see Figure A1),

the water storage variation for soil water models, i.e., small scale water yield $\widetilde{S_y}$, is given by

$$\widetilde{S_y} = \frac{[\omega(T^{J+1}) - \omega(T^J) + \theta_s \cdot \Delta z_t]}{\Delta z_t} + \mu_s \cdot \overline{M} \tag{A1}$$

where $\omega$ is the vadose zone water amount in the moving balance domain, see Figure A2b, $\omega(t) = \int_{z_t(t)}^{z_s} \theta(t, z)\, dz$; $\Delta z_t = \sum_{j=1}^{N} dz_t^j = z_t(T^{J+1}) - z_t(T^J)$ is the total fluctuation of phreatic surface during time period $\Delta T = \sum_{j=1}^{N} dt^j = T^{J+1} - T^J$, and $\theta_s$ is the saturated soil water content. The small-scale water balance in the moving balancing domain at time t is expressed as

$$\left[q_{top} + l \cdot dz_t/2 - q_{bot}\right] \cdot dt = \omega(t) - \omega(t - dt) + \theta_s \cdot dz_t \tag{A2}$$

where $q_{top}$ $(= K(h) \cdot \partial(h + z)/\partial z|_{z=z_s})$ and $q_{bot}$ $(= K(h) \cdot \partial(h + z)/\partial z|_{z=z_b})$ are the nodal fluxes into and out of the moving balancing domain, a fixed top boundary $z_s$ and the moving bottom boundary $z_b = \min(z_t(t), z_t(t - dt))$. $dz_t = z_t(t) - z_t(t - dt)$ is the fluctuation of the phreatic surface during dt, and $l$ is the saturated lateral water flux into the moving domain at time $t$.

Temporally integrating Eq. A2 from time $T^J$ to $T^{J+1}$, we have the macro-time scale water balance as

$$R_{top} + \varepsilon_l - R_{bot} = \omega(T^{J+1}) - \omega(T^J) + \theta_s \cdot \Delta z_t \tag{A3}$$

where $R_{top}$ is the cumulative water flux at $z_s$. $R_{bot}$ is the cumulative water flux out of the moving domain, and $\varepsilon_l$ is the cumulative lateral water flux into the moving balance domain, which can be neglected from the small-scale water balance analysis.

Rephrasing Eq. A1 and A3, the small-scale specific yield $\widetilde{S_y}$ is expressed as

$$\widetilde{S_y} = (R_{top} + \varepsilon_l - R_{bot})/\Delta z_t + \mu_s \cdot \overline{M} \tag{A4}$$

Linking with the large-scale specific yield, the upper boundary water flux can be calculated as $F_{top} = $

$[R_{top} + (\overline{S_y} - \widetilde{S_y})\Delta z_t]/\Delta T$. In such way, the large-scale properties in the groundwater models are physically maintained.



**A2. The moving Dirichlet lower boundary**

The bottom node of the soil column is adaptively located at the phreatic surface, which is numerically realized as the area averaged moving Dirichlet boundary

$$z_t(T^J) = \int_{s\in\Pi} H(T)ds / \int_{s\in\Pi} ds \qquad (A5)$$

where is the elevation of water table, is the influencing domain of a soil column, H(T) is potentiometric head solution, as well as the elevation of the phreatic surface, and s is the horizontal area.

To estimate the lower boundary head of a soil column, the linear extrapolation is adopted to reduce the coupling errors and speed up the convergence. The small-scale lower boundary head at time $t$ ($T^J < t \leq T^{J+1}$) is given by

$$z_t(t) = \frac{(t - T^{J-1}) \cdot z_t(T^J) - (t - T^J) \cdot z_t(T^{J-1})}{T^J - T^{J-1}} \qquad (A6)$$

**A3. The Neumann upper boundary**

The governing equation of the activated layer is expressed as

$$\overline{S_y}\frac{\partial H}{\partial t} = \frac{\partial}{\partial x}\left(K\overline{M}\frac{\partial H}{\partial x}\right) + \frac{\partial}{\partial y}\left(K\overline{M}\frac{\partial H}{\partial y}\right) + F_{top} - F_{base} \qquad (A7)$$

where $\overline{M}(= z_s - z_0)$ is the thickness of the phreatic layer, which is defined as the layer below the vadose zone. $z_0$ is the bottom elevation of the top phreatic layer, $z_0 \ll z_s$. $F_{top}$ is the groundwater recharge into the activated top layer of the phreatic aquifer, $F_{top} = K \cdot \partial H/\partial z|_{z=z_s}$. $F_{base}$ is the water release into the

underlying numerical layer, $F_{base} = K \cdot \partial H/\partial z|_{z=z_0}$. The regional-scale specific yield $\overline{S_y}$, caused by the fluctuation of the water table, is given by

$$\overline{S_y} = V_W/(A \cdot \Delta H) \qquad (A8)$$

where $V_W$ is the amount of water release by the fluctuation of the phreatic surface $\Delta H$, and $A$ is the area of interest.



**Tables and Figures**



**Table 1. Statistical performance of used models in terms of the spatial variation of groundwater table levels.**

| Models | $R^2$ | RMSE (m) | MAE (m) | Remarks |
|---|---|---|---|---|
| MODFLOW | 0.9996 | 0.138 | 0.17 | Calibration (1979-2018) |
| MODFLOW | 0.9987 | 0.203 | 0.15 | |
| HYDRUS-MODFLOW | 0.9811 | 0.946 | 0.754 | Simulation (2016-2018) |
| STEMMUS-MODFLOW | 0.9727 | 0.861 | 0.631 | |

Note: $R^2 = 1 - \frac{\sum_{i=1}^{n}(y_i - \widehat{y_i})^2}{\sum_{i=1}^{n}(y_i - \bar{y})^2}$, $MAE = \frac{\sum_{i=1}^{n}|y_i - \widehat{y_i}|}{n}$, $RMSE = \sqrt{\frac{\sum_{i=1}^{n}(y_i - \widehat{y_i})^2}{n}}$, where $y_i$, $\widehat{y_i}$, are the measured and

model simulated values of the groundwater table levels; $\bar{y}$ is the mean values of the measurements; $n$ is the

number of data points.





**Table A1. The soil hydraulic parameters used in the test cases.**

| Cases | Soil type | $\theta_r$ (cm$^3$ cm$^{-3}$) | $\theta_s$ (cm$^3$ cm$^{-3}$) | $\alpha$ (cm$^{-1}$) | $n$ | $K$s (cm d$^{-1}$) |
|-------|-----------|------|------|-------|------|-------|
| Case 2D | Loam | 0.078 | 0.43 | 0.036 | 1.56 | 24.96 |
| Case 3D | Sandy loam | 0.065 | 0.41 | 0.075 | 1.89 | 106.1 |



**Table A2. A brief summary of current research on soil water-groundwater (SW-GW) coupling**

| Code | GWSiB | | CLM-Parflow | | CATHY/NoahMP | | CP v1.0 | | ELCIRC-SUTRA | | SWAT-MODFLOW | | SWAP-MODFLOW | | HYDMOD | | LEAF2-Hydro | PCR-GLOBWB-MODFLOW | |
|---|---|---|---|---|---|---|---|---|---|---|---|---|---|---|---|---|---|---|---|
| | SiB2 | AquiferFlow | CLM | Parflow | NoahMP | CATHY | CLM4.5 | PFLOTRAN | ELCIRC | SUTRA | SWAT | MODFLOW | SWAP | MODFLOW | HYDRUS | MODFLOW | LEAF2-Hydro | PCR-GLOBWB | MODFLOW |
| Coupling method | replacing the three-layer soil moisture simulation in the SiB2 by the unsaturated zone water movement simulation of AquiferFlow | | replacing the soil column/root-zone soil moisture formulation in CLM with the ParFlow formulation | | Exchange boundary information | | PFLOTRAN's RICHARDS mode to replace CLM4.5's flow model | | Exchange boundary information | | Exchange boundary information | | Exchange boundary information | | Exchange boundary information | | Water table was used as the bottom boundary of soil column, recharge flux was calculated to resolve the water table | One way coupling, net recharge and river levels from PCR-GLOBWB are used as input for MODFLOW's recharge and river, drain packages. | |
| Temporal coupling | two alternative schemes: a. the same temporal resolution; b. use the daily timestep for AquiferFlow and hourly timestep for SiB2 | | same temporal resolution | | CATHY's varing time step adapted to match NoahMP time steps | | same temporal resolution | | run in parallel using time steps of different sizes | | same temporal resolution | | different timesteps, exchange the averaged water table depth and net recharge flux in each MODFLOW timestep | | different timesteps, exchange the averaged water table depth and net recharge flux in each MODFLOW timestep | | same temporal resolution | daily timestep runs for PCR-GLOBWB, pass the monthly average outputs to MODFLOW | |
| Spatial coupling | assign each AquiferFlow grid with one SiB2 soil column | | same spatial resolution | | variables transform from cells to nodes, vice versa | | sparse matrix vector multiplication to interpolate griddes data from one model to the other | | compatible unstructured meshes | | HRU to cell conversion | | each SWAP zone consists one or more MODFLOW grids | | each HYDRUS zone consists one or more MODFLOW grids | | same spatial resolution (1.25 km) | same spatial resolution (5') | |
| Water/Heat capability | water and water energy flow | | water and water energy flow | | yes | only water flow | yes | yes, not considered | water flow | yes, not considered | water and heat flow | | yes, neglect here | water flow | yes, neglect here | water flow | water flow | water flow | Water flow |
| Hydraulic conductivity | calibrated against GWT observations | | Gleeson et al. (2011) permeability maps | | calibrated against discharge observations | | from Williams et al. (2008) | | experimental cases | | from the National Groundwater Information Management and Service Center | | pumping tests | | experimental cases | | from soil database (FAO, 1974) | Gleeson et al. (2011) permeability map | |
| Aquifer thickness | borehole logging data | | Assumed 100 m | | assumed 4 m as the bottom boundary | | assumed 32 m | | experimental cases | | calibrated against the total stream flow | | Borehole data | | experimental cases | | Thickness represented by e-folding depth, which is function of slope | Statistical method for thickness based on topography | |
| Reference | Tian et al. (2012) | | Maxwell and Miller (2005) | | Niu et al. (2011, 2014), Camporese et al. (2010) | | Bisht et al. (2017) | | Yuan et al. (2011) | | Kim et al. (2008) | | Xu et al. (2012) | | Seo et al. (2007) | | (Fan et al., 2007; Miguez-Macho et al., 2007; Fan et al., 2013) | de Graaf et al. (2017) | |

Note: GWT: groundwater table depth, HRUs: Hydraulic response units.



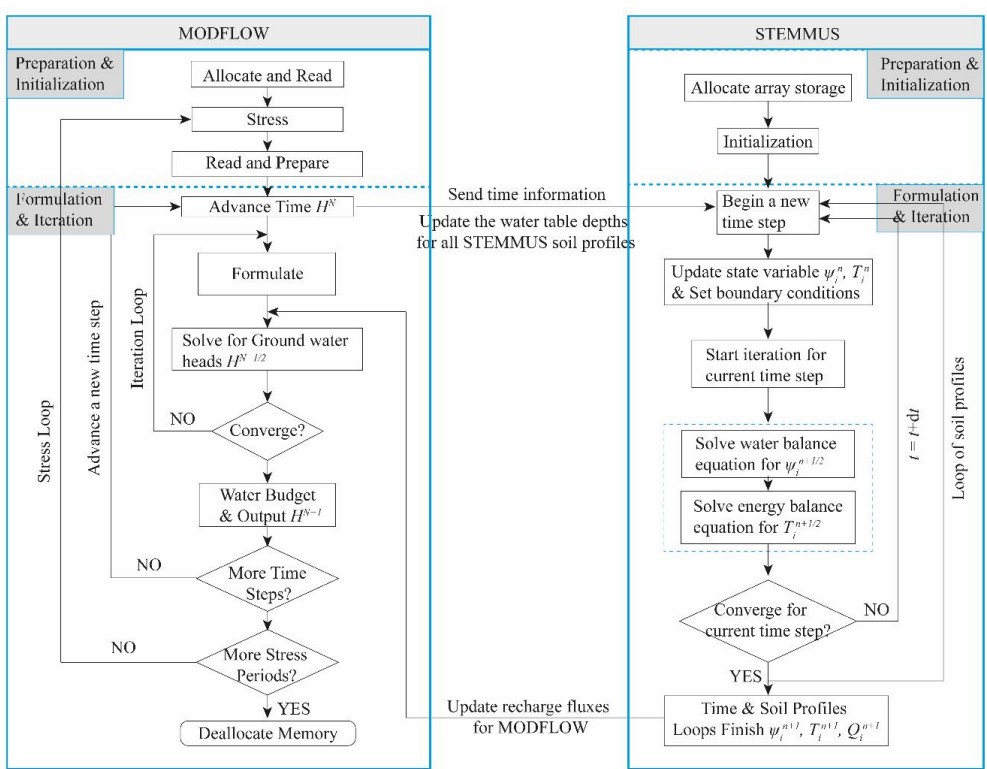

**Figure 1. The schematic diagram for the coupling procedure of soil water model STEMMUS and groundwater model MODFLOW. $H$ is the updated ground water head. $\psi$ is the soil matric potential, $T$ is the soil temperature, and $Q$ is the net recharge fluxes. $n$, $N$ are the current timestep for STEMMUS and MODFLOW, respectively. $n+1$, $N+1$ are the next timestep for STEMMUS and MODFLOW, respectively. $n+1/2$, $N+1/2$ represent for the intermediated state for STEMMUS and MODFLOW, respectively. $i$ is the $i$th soil column.**





(a) Spatial coupling

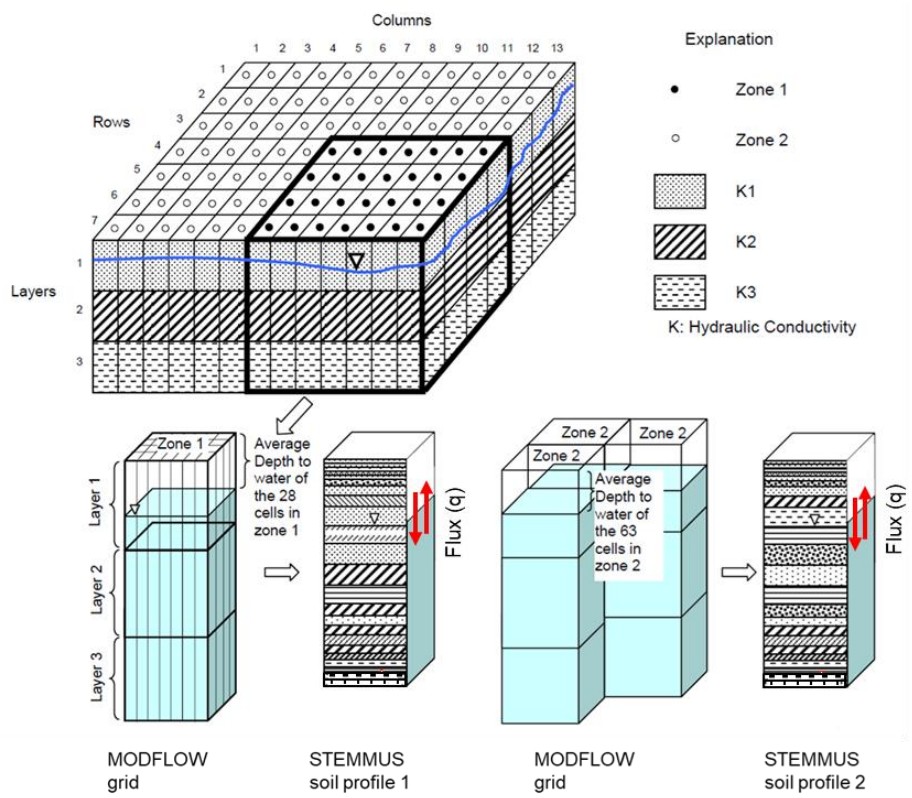

(b) Temporal coupling

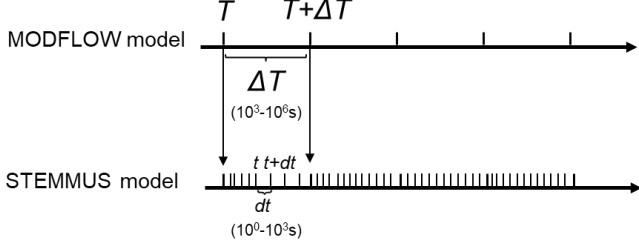

**Figure 2. The schematic diagram for the (a) spatial coupling and (b) temporal coupling of STEMMUS and MODFLOW, adapted from (Seo et al., 2007).**





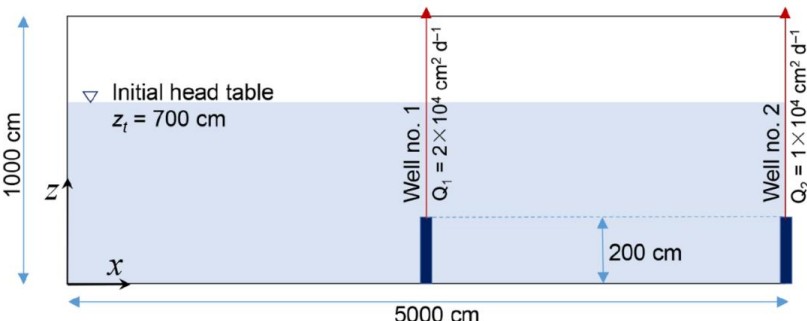

**Figure 3. Schematic of the cross section for test case 2D. Two pumping wells with screens of $z = 0$–200 cm is located at $x = 2500$ and 5000 cm. The pumping rates per unit width at well no. 1 and no. 2 are $2\times10^4$ and $1\times10^4$ cm$^2$ d$^{-1}$, respectively. Adapted from (Zeng et al., 2019).**



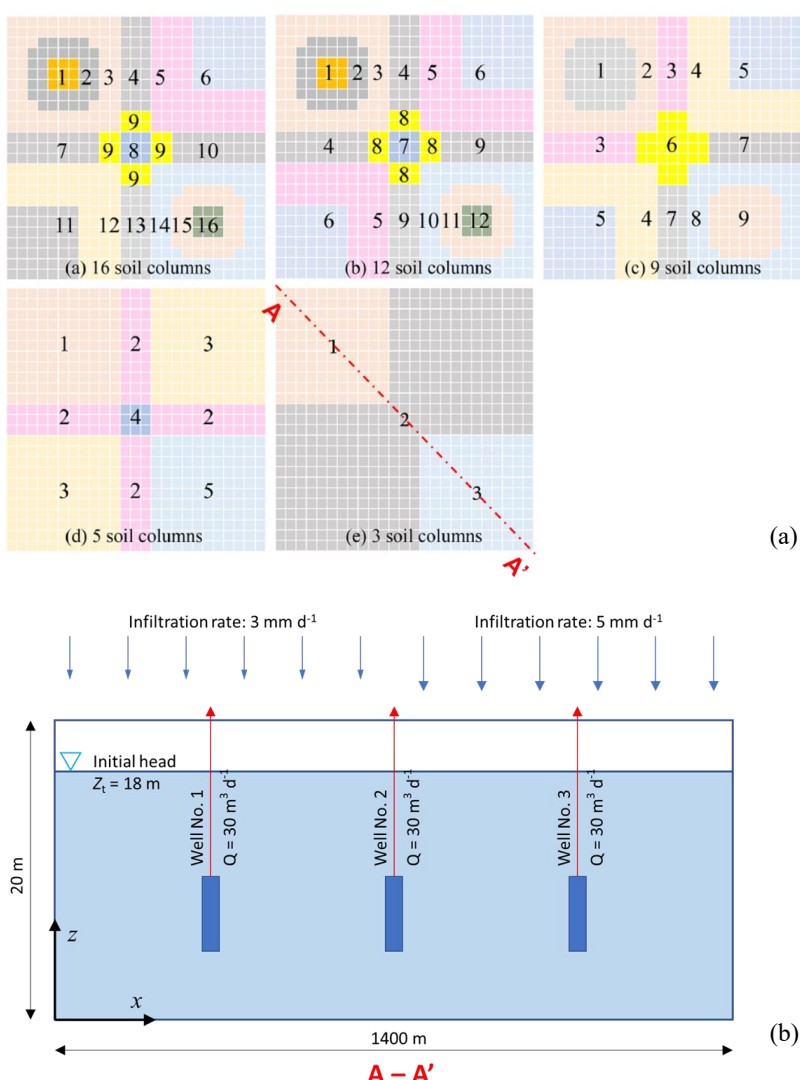

**Figure 4.** Characterization of (a) subzones partitioned for the quasi-3D simulations and (b) cross section A–A' in case 3D. The vadose zone is partitioned into 16, 12, 9, 5, and 3 subzones. Adapted from Zeng et al. (2019).





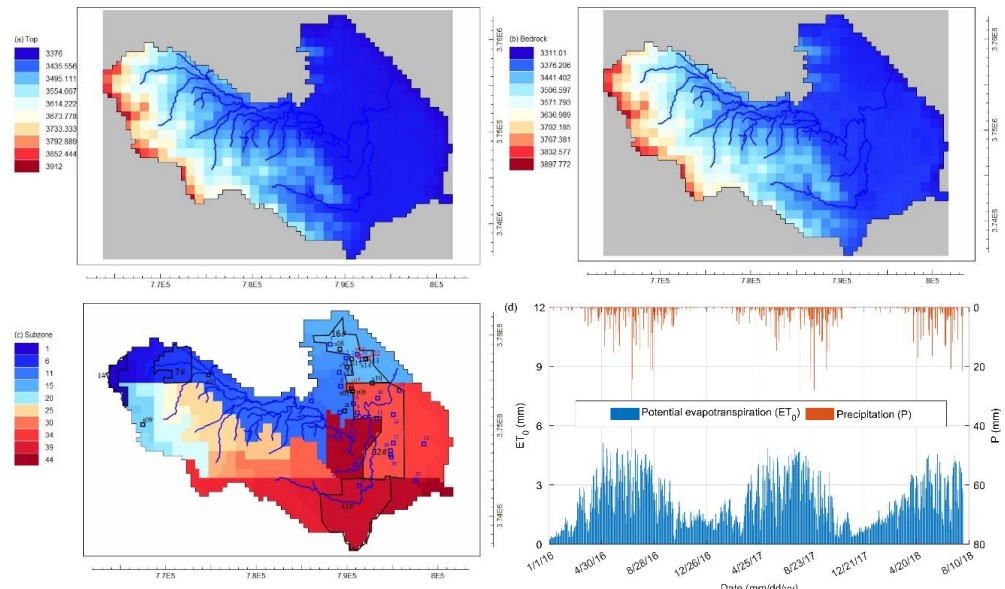

**Figure 5. (a) Land surface elevation, (b) bedrock elevation of phreatic aquifer, (c) subzones of soil**

**columns for Maqu catchment problem, and (d) Spatial averaged daily meteorological forcing**
**(potential evapotranspiration ET$_0$ and precipitation P) in the Maqu catchment. Blue line is the river**
**network. Label number with 's' indicates the soil moisture and temperature profile measurements,**
**the red ones are the current available observation points. Blue square dots indicate the position of the**
**groundwater table level measurements on 5 August 2018. Unit is in m for a, b.**





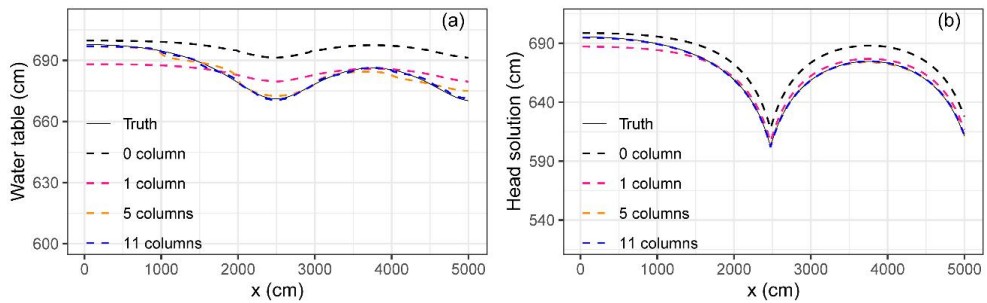


**Figure 6.** Comparison of water table (estimated by fully 3D VSF, MOFLOW only, and STEMMUS-MODFLOW model with 1, 5, 11 soil columns) and head solution (z=0) (estimated by fully 3D VSF, MOFLOW only, and STEMMUS-MODFLOW model with 1, 5, 11 soil columns). Truth is achieved using the fully 3D VSF model. 0 column indicates the MODFLOW only simulations. The moving

boundary method is used in the STEMMUS-MODFLOW model in Case 2D.

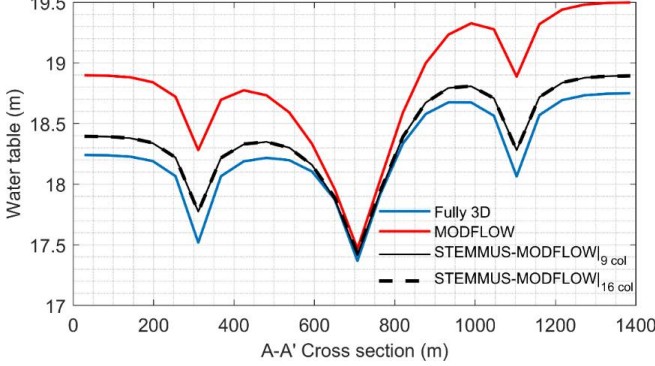

**Figure 7.** Comparison of the phreatic surface at A-A' cross section estimated by the fully 3D Model VSF, MODFLOW only, and the STEMMUS-MODFLOW model with 9 and 16 soil columns, respectively.

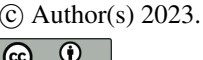

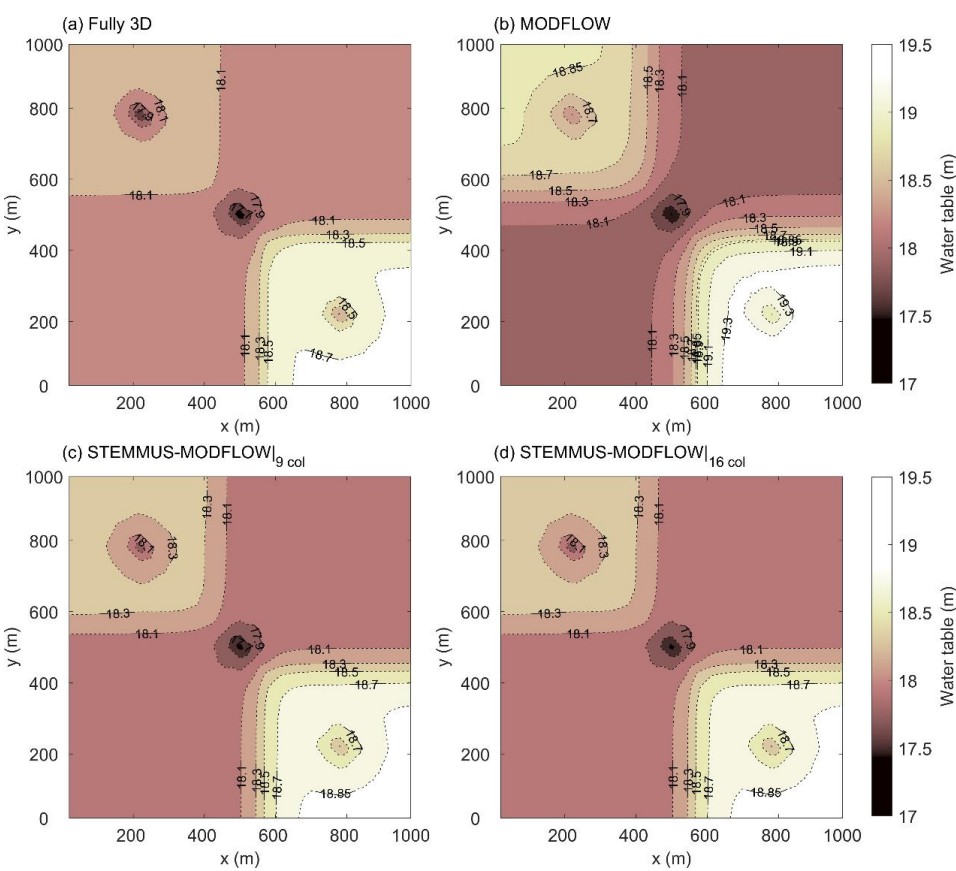

**Figure 8. Comparison of contours of the phreatic surface solution with (a) the fully 3D VSF model, (b) the MODFLOW only, (c) the STEMMUS-MODFLOW model with 9 soil columns, and (d) the STEMMUS-MODFLOW model with 16 soil columns.**



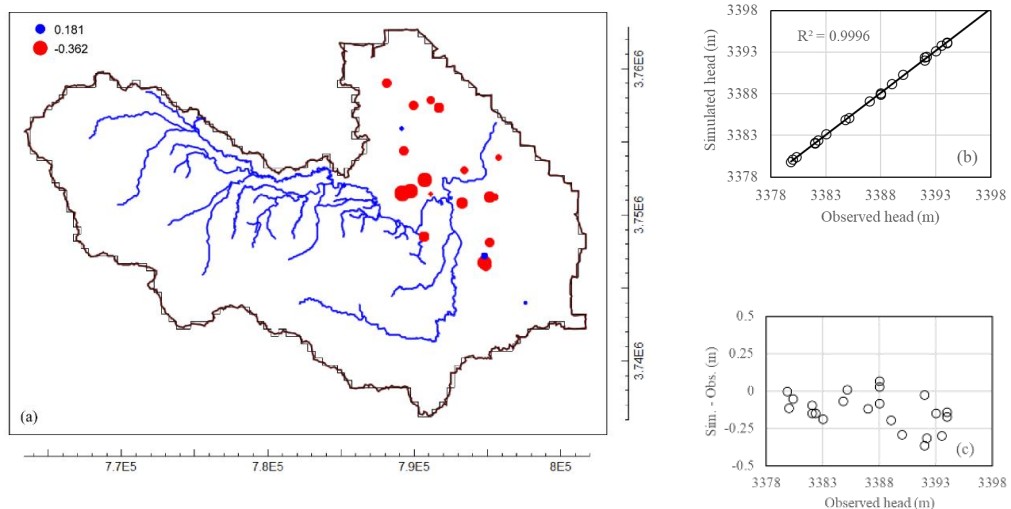

**Figure 9. Model calibration results using MODFLOW for the long-term period against the water table observations of 5 August 2018: (a) spatial distribution of the relative errors, (b) correlation, and (c) the difference between the observed and MODFLOW simulated hydraulic head. Unit is in m.**


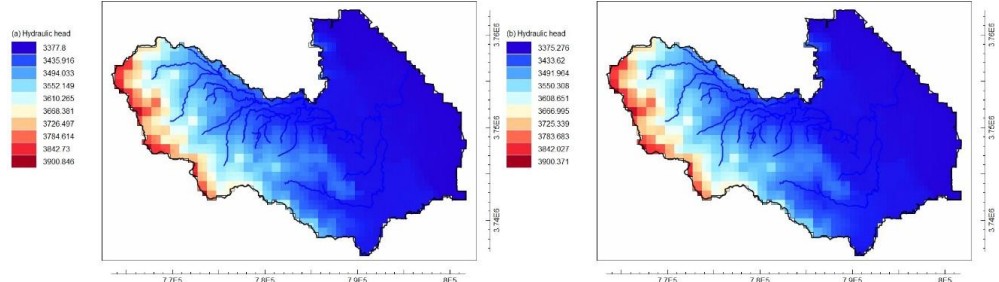

**Figure 10. Comparison of elevation of hydraulic head solutions (the bottom layer) estimated by (a) MODFLOW and (b) the STEMMUS-MODFLOW, on 5 August 2018. Blue line is the river network.**


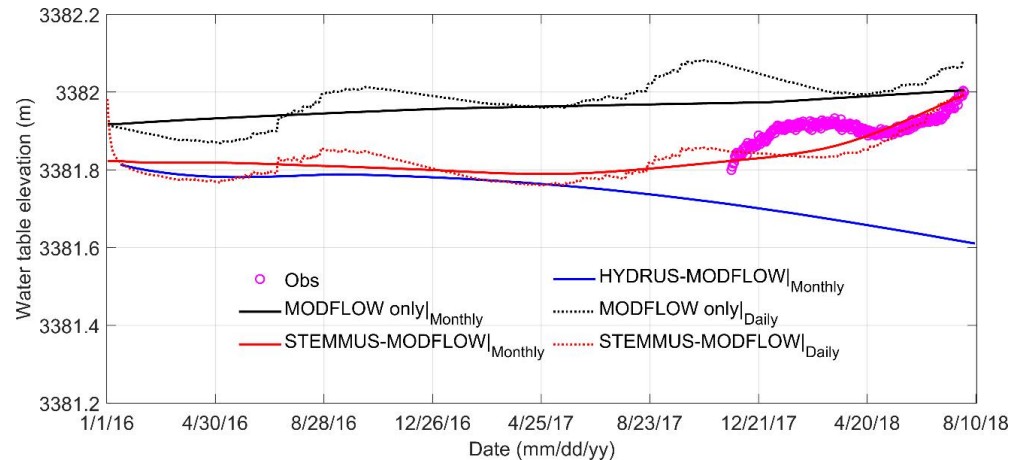


**Figure 11. Comparison of the observed and model estimated water table elevation dynamics using MODFLOW only, HYDRUS-MODFLOW, and STEMMUS-MODFLOW. The observed groundwater table levels are from the installed monitoring well (No. 2 well as shown in Figure 5c). Note that the daily simulations using HYDRUS-MODFLOW was not shown due to numerical**

**instability.**

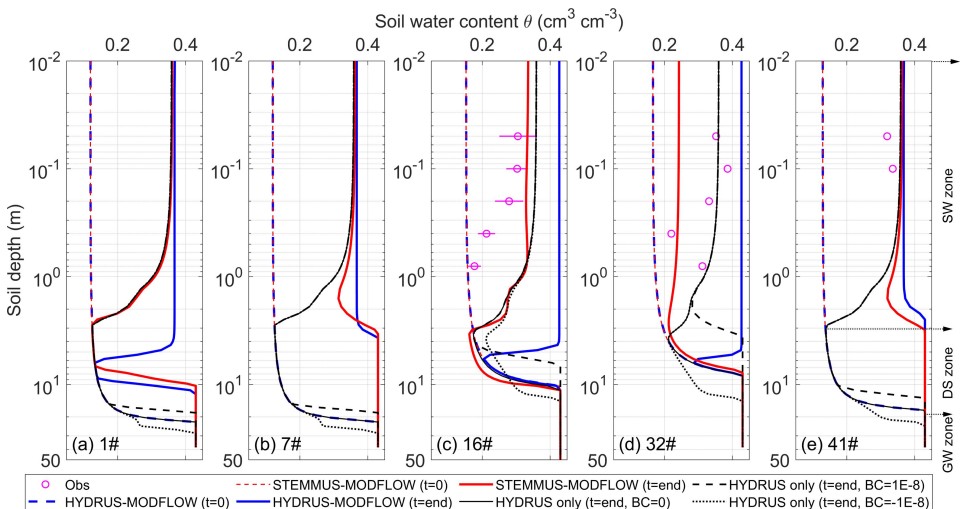

**Figure 12. Comparison of soil water content profiles at the end-time (Aug. 5 2018), of subzones (a) 1#, (b) 7#, (c) 16#, (d) 32#, (e) 41#, as shown in Figure 5c, from the soil moisture measurements (Obs) and estimated by HYDRUS-MODFLOW and the STEMMUS-MODFLOW at the start and end time. HYDRUS model (without MODFLOW) simulations together with different bottom boundary fluxes (BC = 0, 1E-8, and -1E-8 m s⁻¹) were used as reference. GW, DS, and SW zone represents groundwater, deep soil, and soil water zone. The exemplary zone division is based on HYDRUS only simulation (t = 0, BC = 0 m s⁻¹).**


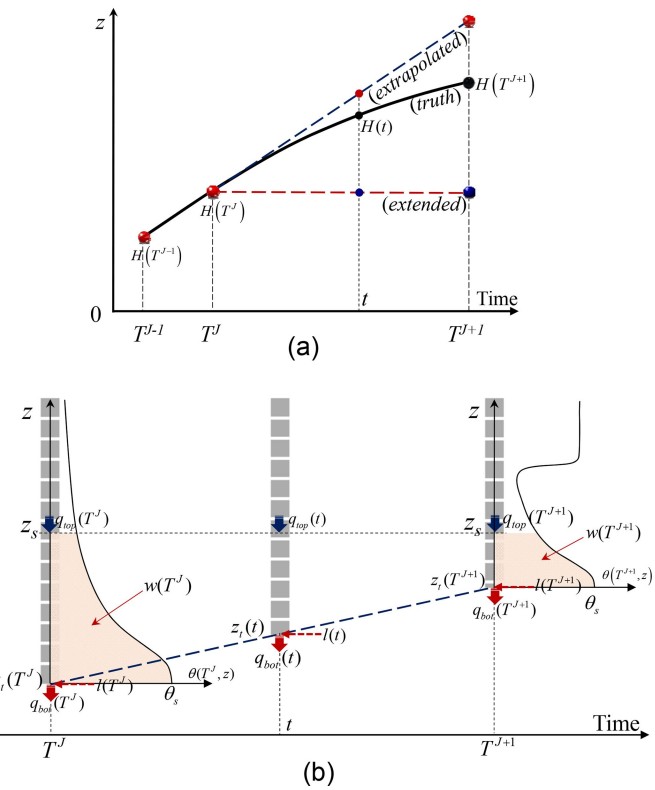

**Figure A1. The Dirichlet–Neumann coupling of the soil-water and groundwater flow models at different scales. (a) Linear or stepwise prediction of Dirichlet lower boundary for the soil-water flow model. (b) Water balance analysis based on a balancing domain with moving lower boundary. Blue dashed line is the linearly extrapolated groundwater table as an alternative for prediction of Dirichlet lower boundary. J (or j), T (or t), and ΔT (or dt) are the time level, time, and time-step size at coarse (or fine) scale. At any of the transient states (t), the balancing domain is bounded by a user-specified top elevation ($z_s$) and the moving phreatic surface ($z_t$). At a transient time t (or $T^J$), the total mass volume in the moving balancing domain is indicated by $\omega(t)$ (or $\omega(T^J)$). The saturated lateral flux of the moving domain is indicated by $l(t)$, while the unsaturated lateral flux is neglected as the assumption of quasi-3D models. The water flux into and out of the balancing domain is indicated by $q_{top}$ and $q_{bot}$. Figure adapted from Zeng et al. (2019).**