# Peer review of "STEMMUS-MODFLOW v1.0.0: Integrated Understanding of Soil Water and Groundwater Flow Processes: Case Study of the Maqu Catchment, north-eastern Tibetan Plateau"

_Geoscientific Model Development, 2022_

## Author Comment (AC1)

We thank the editor and reviewers very much for the dedicated time and efforts they put to improve this manuscript with constructive comments. We made our point-by-point response in blue fonts as below. The referee comments are in black fonts.

**RC1**: Referee comments from #1 anonymous referee.

**RC2**: Referee comments from #2 anonymous referee.

Anonymous Referee #1

The presentation of the aims and of the methods are not clear enough to allow a proper review. The english level should also be improved. Thus I recommend to reject this manuscript, and to encourage the authors to resubmit after a thorough rewritting step.

**Response:** We thank the reviewer for the critical comments. The aim/main interest of this manuscript is to develop and validate a soil water-groundwater model, STEMMUS-MODFLOW, using a two-way feedback coupling scheme. STEMMUS-MODFLOW integrated the commonly used groundwater model MODFLOW with the physically based unsaturated soil water model STEMMUS. The coupled model has the potential to further enhance its physical interpretations regarding the frozen soil (STEMMUS-FT), snow cover (STEMMUS-UEB), ecohydrology (STEMMUS-T&C), etc. Furthermore, to our knowledge, this study is the first over Tibetan plateau using the model with soil water-groundwater coupling considering two-way feedbacks.

We clarify further the method in the updated Section 2 and Appendix A.

We also polish the English writing as suggested.

**General comments :**

[1] - The abstract is unclear, it should allow to understand what in this work is compared : coupling approaches or simulation softwares, or both. If comparisons of both approaches and softwares are made, they should be articulated with each other (different coupling approaches may be implemented using either of the considered softwares).

**Response:** We intended to compare different coupling approaches, i.e., no soil water-groundwater coupling (only MODFLOW), weakly two-way coupling (HYDRUS-MODFLOW), two-way feedback coupling (STEMMUS-MODFLOW), and the full soil water-groundwater coupling (MODFLOW-VSF) in this work.

We rewrite the abstract and make the aim of this manuscript clear, and further add a table (Table R1) to illustrate the difference in the coupling approaches between the used software in this manuscript.

Table R1. A comparison of soil water-groundwater (SW-GW) coupling approaches used in the numerical experiments.

| Numerical Experiments | SW-GW coupling approaches | Temporal coupling | Spatial coupling | Reference |
|---|---|---|---|---|
| MODFLOW | no coupling | / | / | Harbaugh, 2005 |
| HYDRUS-MODFLOW | weakly coupled, exchange boundary information | different time steps, exchange the averaged water table depth and net recharge flux in each MODFLOW time step | each HYDRUS zone consists one or more MODFLOW grids | Seo et al., 2007 |
| STEMMUS-MODFLOW | two-way feedback coupling, iterative feedback coupling scheme, moving lower boundary approach for soil water model | different time steps, exchange the averaged water table depth and net recharge flux in each MODFLOW time step | each STEMMUS zone consists one or more MODFLOW grids | this study |

| | fully coupling, solves the three-dimensional form of the Richards equation for the entire MODFLOW domain | | | |
|---|---|---|---|---|
| MODFLOW-VSF | fully coupling, solves the three-dimensional form of the Richards equation for the entire MODFLOW domain | smaller time steps than MODFLOW | same spatial resolution | Thoms et al., 2006 |

**[2]** - To the knowledge of the reviewer, an important example of hydrological model that couples dimensionnaly heterogeneous descriptions of flow in the saturated zone and in the unsaturated zone is MIKE-SHE (e.g.: Graham and Butts, 2005), which is for instance included in recent international benchmarking efforts for physically based hydrological modeling (e.g.: Kollet *et al.*, 2016). The fact that works related to MIKE-SHE do not appear in the references of the manuscript make me think that the bibliographical survey on which the presentation of the background of the study is done should be consolidated.

**Response:** We agree that MIKE-SHE is a widely used model solving the coupled unsaturated and saturated flow processes and include references of MIKE-SHE in our introduction. Nevertheless, its coupling strategy is similar to the models we already referred to (no overlap of the two compartments, compartments solved separately, iterative procedure with step-wise adjustment of water table to improve mass balance).

We also add some other relevant references in the Introduction, regarding the groundwater modelling efforts (Maxwell et al., 2014; Kollet et al., 2016; Grenier et al., 2018).

**[3]** - Since I did not understand the interest that the authors see for the proposed approach (I am not saying that there is no interest ; it is just not stated), and since I did not understand the approach itself, I stop my review at the section 2.2.4. Obviously in such circumstances I will not be able to review the application cases and their discussions.

**Response:** The interest of the adopted two-way feedback coupling approach in STEMMUS-MODFLOW is that it sustains the physical reality of the unsaturated-saturated processes and is computational efficient. In addition, using the multi-scale water balance analysis, it solves

the scale-mismatch problem, which commonly exists in the traditional two-way coupling scheme. Furthermore, using the moving lower boundary strategy, the coupled model also reduces the potential coupling errors induced by the lateral water fluxes.

We add further explanations regarding the proposed approaches in Section 2 and Appendix A.

The benefit of the adopted approaches is illustrated as below:

**a. merits of the multi-scale water balance analysis**

Figure R1 shows the dynamics of water table for the 1D case in Zeng et al. (2019), with constant upper boundary, from different numerical solutions with various macro-time-step sizes. In which, 'Truth' is the HYDRUS-1D solution. 'Stepwise' is the solution from Seo et al. (2017), no multi-scale water balance analysis. 'Linear' is the numerical solution from Zeng et al. (2019), which conducted the multi-scale water balance analysis.

From Figure R1, the disagreements between the simulations and the 'Truth' increased, especially for the numerical solutions from Seo et al. (2007). In contrast, the linear prediction, which adopted the multi-scale water balance analysis, agrees well with the 'Truth'. It clearly indicates that the SW-GW coupling errors commonly exists in the traditional non-iterative feedback coupling methods (Seo et al., 2007), which increases with the macro-time-step sizes. Such SW-GW coupling errors can be significantly reduced by the numerical solution with the multi-scale water balance analysis.

[Figure]

**Figure R1**. Water table changing with time for different macro-time-step sizes (1T D 0:005, 0.05, 0.1, and 0.2 days), in scenario 1, case 1. From Zeng et al. (2019).

The HYDRUS-1D solution is taken as the truth. Compared with the stepwise extended method (Seo et al., 2007), the coupling error is significantly reduced by a linear prediction.

**b. benefits of the moving lower boundary approach**

Figure R2 presents the comparison of the cross-sectional RMSE of the phreatic surface and the head solution at the bottom layer between the models with the moving lower boundary and stationary-lower boundary methods. The coupling errors (RMSE) from the stationary lower boundary methods were significantly larger than that from the moving lower boundary method. For the stationary lower boundary method, the RMSE decreased with the decrease of soil column length (i.e., the length of the saturated soil column). It is induced by the nontrivial lateral water fluxes between adjacent saturated soil columns of the stationary lower boundary method.

[Figure]

**Figure R2**. Comparison of RMSE of (a) the phreatic surface and (b) the head solution (at *z* = 0) between the moving-boundary and the stationary-boundary methods. Three different lengths of the stationary soil columns, L = 1000, 500, and 300 cm, are considered. From Zeng et al. (2019).

Abstract:

[4] l 22 « and physically » clumsy expression (also l 27 « proven physically accurate »)

**Response:**

L 22: "efficiently and physically" is rephrased as "in a computationally-efficient and physically-based manner".

L27: "proven physically accurate" is rephrased as "proven accurate".

**[5]** l 23-24 : « We present … and verified » : time concordance problem

**Response:** Thank you for pointing out this mistake, "present" is changed to "presented"

**[6]** l 32 : « HYDRUS-MODFLOW » : do you mean the variably saturated flow reference soultion, or something else ? If this is, as I think, something else, it should be explicited.

**Response:** HYDRUS-MODFLOW use the weakly two way coupling strategy.

We will add a Table (Table R1) to explicitly explain the coupling physics of MODFLOW, HYDRUS-MODFLOW, STEMMUS-MODFLOW, and MODFLOW-VSF, as in appendix A.

**[7]** l 38-39 : « spatiotemporal heterogeneity of soil water-groundwater interactions » sounds strange.

**Response:** It is rephrased as "heterogeneous descriptions of soil water-groundwater interactions".

**[8]** L 43 : "from bedrock to atmosphere" Why is there quotation marks here ? Are you citing some one ?

**Response:** The quotation marks will be removed. "from bedrock to atmosphere" is rephrased as bedrock-to-atmosphere.

Introduction:

**[9]** l 115 : « In this study, we coupled the soil water model (STEMMUS) with the groundwater model (MODFLOW) in a two-way feedback manner. ». The authors should explicitly state here why did they choose these softwares and this coupling method. What is the added value compared to the existing literature ?

**Response:**

MODFLOW is a modularized groundwater flow model, developed by the U.S. Geological Survey, which is considered an international standard for groundwater modeling and one of the most commonly used groundwater software. It well represents the physical processes

related to groundwater flow, including recharge, discharge, wells, drawdown, etc. MODFLOW and related tools bring the capability of addressing the current and future challenges to groundwater resources.

STEMMUS is a detailed, physically based two-phase flow soil model. It was first developed to investigate the underlying physics of soil water, vapor, and dry air transfer mechanisms and their interaction with the atmosphere (Zeng et al., 2011a, b; Zeng and Su, 2013). It is achieved by simultaneously solving the balance equations of soil mass, energy, and dry air in a fully coupled way. Recently, STEMMUS is developed to represent the following processes, root water uptake (Yu et al., 2016), freeze-thaw process (Yu et al., 2018; Yu et al., 2020b), snow accumulation and melt process (Yu et al., 2021), soil and vegetation biogeochemical process (Wang et al., 2021; Yu et al., 2020a). It facilitates our understanding of the hydrothermal dynamics of soil-plant-atmosphere continuum (SPAC).

Compared with the existing literature, the adopted two-way feedback coupling method, implemented with the moving lower boundary and multi-scale water balance strategy, can improve the numerical stability and reduce the coupling errors.

We add the explanations why choose MODFLOW, STEMMUS and the coupling approaches and their added values here.

**[10]** L 120 : section 2 should be splitted, with a dedicated section for the governing equations and the resolution methodologies, and another section dedicated to the set up of the test cases.

**Response:** Section 2 is restructured into two sub-sections. Section 2.1 is dedicated to the description of the coupled STEMMUS-MODFLOW. The description of governing equations will be elaborated in Section 2.1.1. Section 2.1.2 will explain the coupling procedure, including the spatial coupling and temporal coupling methodologies, and the multi-scale interface water balance analysis. Section 2.2 will be dedicated to the setup of the test cases, including the 2D case (large groundwater drawdown) and 3D case (pumping and irrigation), and the regional groundwater case (Maqu catchment and model setup).

Section 2:

**[11]** L 125-126 : « MODFLOW assumes that the ground water is with constant density and the porous medium is noncompressible » : the scientific english here should be improved. « assumes that the density of water is constant and that the porous medium is incompressible » would sounds better I think. Overall I think that the level of english langage should be improved. From now on I stop pointing examples of this kind.

**Response:** Thanks a lot. We rephrase it as "MODFLOW assumes that the density of water is constant and that the porous medium is incompressible.". We will carefully polish the English writing.

**[12]** L 161 : « After a certain number of iterations, STEMMUS model will converge ». What is the involved iterative process ? Is STEMMUS run until equilibrium, is this a fixed point method, is this something else ? This key point should be explained in detail.

**Response:**

The Picard iterative solution strategy is implemented in STEMMUS as following steps:

1) Start with an initial estimate of $\psi^k$ and $T^k$. Extrapolate the solutions for the last two time steps (k-2 and k-1) forward to obtain an estimate of $\psi^k$ and $T^k$ (If this is the first time step, only the k -1 'solution' is available - it is the initial condition. In this case, we can assume $\psi^k$ and $T^k$ are given by the initial conditions as the first guess);

2) Use the estimated states to evaluate all components (except $\psi^k$) in soil water equation. Solve the resulting tridiagonal matrix equation for $\psi^k$;

3) Use the latest estimates of $\psi^k$ and $T^k$ to evaluate and solve for $T^k$ in soil energy equation;

4) Repeat steps 2 and 3 until the convergence criterion, i.e., the difference between successive estimates becomes smaller than some predetermined tolerance level, is met.

With this algorithm, we never need to solve a matrix equation any more complex than a tridiagonal matrix. A very fast procedure exists for the solution of such equations (e.g. Thomas algorithm).

We add the key points of the iterative process in Section 2.1.2.

**[13]** L 156 Figure 1 : The figure is too rich with both informatic (e.g. : allocate array storage) and algorithmic (e.g. : Solve equation, converge yes/no …) for the two models, and unsufficiently explained. For instance what is the 'stress periods' in MODFLOW ? What is the loop of soil profil in STEMMUS ? Of course this very important part of the paper should be as complete and informative as possible, and this requires more explanations.

**Response:**

We add more explanations about Figure 1 (in Section 2.1.2).

"Allocate array storage", prior to STEMMUS simulations, the memory is allocated for the used array in STEMMUS. These array include the following: time and vertical domain information, initial soil matric potential and temperature information, soil properties, meteorological forcing information, and control information for the iterative solution algorithm.

"Solve equation, converge yes/no", STEMMUS model use the iterative solution to solve soil water balance and energy balance equation simultaneously (see for the iterative solution). When the convergence criterion is achieved, STEMMUS succeed to update soil matric potential and temperature in the current time step. If not, STEMMUS will continue running.

What is the 'stress periods' in MODFLOW?

The simulation time, in MODFLOW, is divided into stress periods. Stress periods are used to define time intervals during which the external stresses (e.g., pumping, precipitation, etc.) for the MODFLOW remain constant. For example, if the pumping rate on a well were to change on a monthly basis, the stress periods would need to be one month in length or less (Harbaugh, 2005). Stress periods are further broken up into the time steps, which are the potentially non-equidistant discretization of time used to solve the numerical models in the simulation.

What is the loop of soil profile in STEMMUS ?

Soil profiles: when coupling STEMMUS with MODFLOW, the whole simulation domain is divided into several zones for STEMMUS simulations (2 STEMMUS soil profiles as shown in Fig. 2).

The loop of soil profiles means that running STEMMUS circularly (2 times in Fig. 2) to obtain the unsaturated flow simulations for the whole MODFLOW domain. Then update the recharge water fluxes at the bottom of the soil profiles to MODFLOW.

[14] Section 2.2.2 : Hard to follow. The explanation are supposed to be given in the appendix A, which is not understandable (for instance in Figure 2a where are Zs and Zt(t) mentionned in l 728 ?).

**Response:** Section 2.2.2 is dedicated to explain the spatial coupling between STEMMUS and MODFLOW (using Figure 2a as an illustrative example). It includes the areal coupling (horizontal direction), and the coupling between soil water-groundwater interfaces (vertical direction).

For the areal coupling, in Figure 2a, the whole MODFLOW domain was divided into 2 zones, which corresponds to the two STEMMUS soil profiles.

For the vertical direction coupling, the depth to groundwater in each MODFLOW cell of the zone is averaged to determine the hydraulic head at the bottom of the corresponding STEMMUS soil profile. The bottom water flux of STEMMUS soil profile is applied to MODFLOW cell of the corresponding zone as the groundwater recharge.

Zs and Zt(t) are variables used in multi-scale water balance analysis. Zs is the top elevation defined by users and Zt(t) is the moving phreatic surface.

We add the Zs and Zt(t) in Figure 2a (See Figure R3a).

To improve the convergence of the hydraulic head/flux at the phreatic surface, the iterative feedback coupling method was used. We add the explanation of the iterative feedback coupling method in the Appendix A4.

To diminish the effect of the nontrivial lateral fluxes between the saturated regions of the STEMMUS soil profiles, the moving lower boundary method was used. The details were presented in Appendix A2.

(a) Spatial coupling

a, ground surface
b, bottom of STEMMUS soil profile
Depth, depth to ground water

$z_s$, user-specified top elevation
    for the balance domain
$z_t$, the moving phreatic surface

(b) Temporal coupling

[Figure]

**Figure R3**. The schematic diagram for the (a) spatial coupling and (b) temporal coupling of STEMMUS and MODFLOW, adapted from (Seo et al., 2007). $T^J$, $T^{J+1}$, $\varDelta T$ are the macro-time steps in the MODFLOW model, $t$, $dt$ are the micro-time steps in the STEMMUS model.

**[15]** l 180-181 : « Within the given time step, STEMMUS model will adapt its own time step (variable between 10 0 -10 3 s) to the converged simulation results. » Clumsy. I think here the authors are making an assumption of separation of time scales, and this should be clarified and argumented.

**Response:** Yes, we are explaining the difference in the time discretization of MODFLOW and STEMMUS here. We make clarification in the manuscript.

The simulation time in MODFLOW is firstly divided into stress periods, during which the input data for all external stresses are constant. Then, further divided into time steps.

The time step of groundwater models is larger than that of soil water models (Figure R3b). MODFLOW is usually operated with a prescribed time step ($10^3$-$10^6$s).

The time discretization in STEMMUS are designed to be associated with the numerical solution. It starts with a prescribed initial time increment, $\Delta t$. This time increment is automatically adjusted at every time level according to the following rules:

a. The maximum desirable/allowed change of soil matric potential and temperature is set as the criterion for time adjusting.

b. If the changes between two adjacent (time-step-wise) updates of soil matric potential/temperature are less than the maximum change criterion, then STEMMUS continues, the time increment for the next time step is increased by multiplying $\Delta t$ by a constant larger than 1 (usually between 1 and 6).

c. If the changes between two adjacent updates of soil matric potential/temperature are larger than the maximum change criterion, STEMMUS will repeat the current time step with a decreased time increment, multiply by a constant less than 1, which is determined as a ratio between the difference of two adjacent updates of soil matric potential/temperature and the maximum allowed changes.

d. time increments cannot be less than a prescribed minimum time step, $\Delta t_{min}$, nor larger than a maximum time step, $\Delta t_{max}$, (i.e., $\Delta t_{min} \leq \Delta t \leq \Delta t_{max}$).

**[16]** L 189-191 : « By scale matching of the water budget components at the interface of respective scales, water balance is conserved and the upper boundary flux for the groundwater flow model can be achieved. » I don't understand what scale matching means here. In this section once again the authors point the reader to the appendix A, but the term scale matching does not appear in this appendix.

**Response:**

scale-mismatch problem: Using the two-way coupling method, the governing equations and numerical schemes of soil water and groundwater sub-models are built at different scales. For groundwater models (Harbaugh et al., 2005), the specific yield at the phreatic surface is usually represented by a simple large-scale parameter, while for soil-water models (Simunek et al., 2009), the small-scale phreatic water release is influenced by the water-table depth and the unsaturated soil moisture profile (Dettmann and Bechtold, 2016). Delivering small-scale solutions of the soil water models onto the large-scale interfacial boundary of the groundwater model, as well as maintaining the global mass balance, usually introduces significant nonlinearity to the entire SW-GW coupling system (Stoppelenburg et al., 2005).

The mismatch of numerical scales in the coupled soil water and groundwater sub-models causes significant coupling errors and instability. A multi-scale water balance analysis at the phreatic surface (Appendix A1) helps to deal with such problems.

Scale matching here means to link the soil water (small scale) and groundwater (large scale) budget components at the phreatic surface, using the multi-scale water balance analysis.

We add the explanation of the term scale matching. In Appendix A1. Multi-scale water balance analysis, we add the text that it is used to deal with the scale mismatch problems.

Anonymous Referee #2

**General comments.**

**[1] RC2:** The paper aims to develop and validate a soil water-groundwater model, STEMMUS-MODFLOW, that accounts for the vadose zone processes and their interactions with the groundwater flow using a two-way feedback coupling scheme.

**Response:** We thank the reviewer very much for the time and effort and also for the helpful comments.

**[2] RC2:** I appreciate the authors' effort in writing this paper, but I found it hard to read and understand. I think the paper needs to be restructured with a focus on clarity and cohesion and resubmitted. I also noticed that most of the questions that I had from reading the paper were already asked by reviewer 1. I would like the authors to address those questions in addition to improving the writing quality of the paper.

**Response:** We restructure the manuscript, by stating clearly the main interest of this manuscript, adding more explanation of the methods. Please see our response to RC1.

We have the manuscript English edited.

**Reference**

Dettmann, U., and Bechtold, M.: One-dimensional expression to calculate specific yield for shallow groundwater systems with microrelief, Hydrological Processes, 30, 334-340, https://doi.org/10.1002/hyp.10637, 2016.

Grenier, C., Anbergen, H., Bense, V., Chanzy, Q., Coon, E., Collier, N., Costard, F., Ferry, M., Frampton, A., Frederick, J., Gonçalvès, J., Holmén, J., Jost, A., Kokh, S., Kurylyk, B., McKenzie, J., Molson, J., Mouche, E., Orgogozo, L., Pannetier, R., Rivière, A., Roux, N., Rühaak, W., Scheidegger, J., Selroos, J. O., Therrien, R., Vidstrand, P., and Voss, C.: Groundwater flow and heat transport for systems undergoing freeze-thaw: Intercomparison of numerical simulators for 2D test cases, Advances in Water Resources, 114, 196-218, https://doi.org/10.1016/j.advwatres.2018.02.001, 2018.

Harbaugh, A. W.: MODFLOW-2005 : the U.S. Geological Survey modular ground-water model--the ground-water flow process, Report 6-A16, 2005.

Maxwell, R. M., Putti, M., Meyerhoff, S., Delfs, J.-O., Ferguson, I. M., Ivanov, V., Kim, J., Kolditz, O., Kollet, S. J., Kumar, M., Lopez, S., Niu, J., Paniconi, C., Park, Y.-J., Phanikumar, M. S., Shen, C., Sudicky, E. A., and Sulis, M.: Surface-subsurface model intercomparison: A first set of benchmark results to diagnose integrated hydrology and feedbacks, Water Resources Research, 50, 1531-1549, https://doi.org/10.1002/2013WR013725, 2014.

Kollet, S., Sulis, M., Maxwell, R. M., Paniconi, C., Putti, M., Bertoldi, G., Coon, E. T., Cordano,

E., Endrizzi, S., Kikinzon, E., Mouche, E., Mügler, C., Park, Y.-J., Refsgaard, J. C., Stisen, S., and Sudicky, E.: The integrated hydrologic model intercomparison project, IH-MIP2: A second set of benchmark results to diagnose integrated hydrology and feedbacks, Water Resources Research, 53, 867-890, https://doi.org/10.1002/2016WR019191, 2017.

Seo, H. S., Simunek, J., and Poeter, E. P.: Documentation of the HYDRUS Package for MODFLOW-2000, the U.S. Geological Survey Modular Ground-Water Model, Int. Ground Water Modeling Center, Colorado School of Mines, Golden, CO, 96 p., 2007.

Simunek, J., Sejna, M., Saito, H., Sakai, M., and van Genuchten, M. T.: The HYDRUS-1D software package for simulating the one-dimensional movement of water, heat, and multiple solutes in variably-saturated media, Version 4.08, 2009.

Stoppelenburg, F. J., Kovar, K., Pastoors, M. J. H., and Tiktak, A.:Modelling the interactions between transient saturated and unsaturated groundwater flow. Off-line coupling of LGM and SWAP, RIVM Rep., 500026001, 70, 2005.

Thoms, R. B., Johnson, R. L., and Healy, R. W.: User's guide to the Variably Saturated Flow (VSF) process to MODFLOW, Report 6-A18, 58, 2006.

Wang, Y., Zeng, Y., Yu, L., Yang, P., Van Der Tol, C., Yu, Q., Lü, X., Cai, H., and Su, Z.: Integrated modeling of canopy photosynthesis, fluorescence, and the transfer of energy, mass, and momentum in the soil-plant-Atmosphere continuum (STEMMUS-SCOPE v1.0.0), Geosci Model Dev, 14, 1379-1407, https://doi.org/10.5194/gmd-14-1379-2021, 2021.

Yu, L., Zeng, Y., Wen, J., and Su, Z.: Liquid-Vapor-Air Flow in the Frozen Soil, Journal of Geophysical Research: Atmospheres, 123, 7393-7415, https://doi.org/10.1029/2018JD028502, 2018.

Yu, L., Fatichi, S., Zeng, Y., and Su, Z.: The role of vadose zone physics in the ecohydrological response of a Tibetan meadow to freeze–thaw cycles, The Cryosphere, 14, 4653-4673, https://doi.org/10.5194/tc-14-4653-2020, 2020a.

Yu, L., Zeng, Y., and Su, Z.: Understanding the mass, momentum, and energy transfer in the frozen soil with three levels of model complexities, Hydrol. Earth Syst. Sci., 24, 4813-4830, https://doi.org/10.5194/hess-24-4813-2020, 2020b.

Yu, L., Zeng, Y., and Su, Z.: STEMMUS-UEB v1.0.0: integrated modeling of snowpack and soil water and energy transfer with three complexity levels of soil physical processes, Geosci. Model Dev., 14, 7345-7376, https://doi.org/10.5194/gmd-14-7345-2021, 2021.

Zeng, J., Yang, J., Zha, Y., and Shi, L.: Capturing soil-water and groundwater interactions with an iterative feedback coupling scheme: New HYDRUS package for MODFLOW, Hydrol. Earth Syst. Sci., 23, 637-655, https://doi.org/10.5194/hess-23-637-2019, 2019.

Zeng, Y., Su, Z., Wan, L., and Wen, J.: Numerical analysis of air-water-heat flow in unsaturated soil: Is it necessary to consider airflow in land surface models?, Journal of Geophysical Research: Atmospheres, 116, D20107, https://doi.org/10.1029/2011JD015835, 2011a.

Zeng, Y., Su, Z., Wan, L., and Wen, J.: A simulation analysis of the advective effect on evaporation using a two-phase heat and mass flow model, Water Resour. Res., 47, W10529, https://doi.org/10.1029/2011WR010701, 2011b.

Zeng, Y. J., and Su, Z. B.: STEMMUS : Simultaneous Transfer of Engery, Mass and Momentum in Unsaturated Soil, ISBN: 978-90-6164-351-7, University of Twente, Faculty of Geo-Information and Earth Observation (ITC), Enschede, 2013.